# DocOS: Towards Proactive Document-Guided Actions in GUI Agents

**Jingjing Liu** [* 1]  **Ziye Huang** [* 1]  **Zihao Cheng** [* 1]  **Zeming Liu** [1]  **Jiahong Wu** [2]  **Yuhang Guo** [3]  **Kehai Chen** [4]
**Yunhong Wang** [1]  **Haifeng Wang** [5]

## Abstract

While Graphical User Interface (GUI) agents have shown promising performance in automated device interaction, they primarily depend on static parametric knowledge from pre-training or instruction tuning. This reliance fundamentally limits their ability to handle long-tailed tasks that require explicit procedural knowledge absent from model parameters, often forcing agents to resort to inefficient and brittle trial-and-error exploration. To mitigate this limitation, we introduce **Proactive Document-Guided Action** for GUI agents in dynamic, open-web environments, a novel paradigm that mirrors human problem-solving by enabling agents to autonomously search for relevant documentation to resolve long-tailed tasks. To evaluate agents' capability in this paradigm, we propose **DocOS**, a benchmark designed to assess document-guided problem solving in fully interactive environments. DocOS requires agents to autonomously navigate a web browser, locate relevant online documentation, comprehend procedural instructions, and faithfully ground them into executable GUI actions. Extensive experiments reveal that progress is strictly constrained by dual bottlenecks: agents struggle to reliably locate relevant information during proactive search and frequently fail to faithfully ground retrieved instructions into precise actions, pointing toward document-guided interaction as a crucial pathway for enabling self-evolving GUI agents in dynamic environments[1].

[1]School of Computer Science and Engineering, Beihang University, Beijing, China [2]School of Computer Science, Peking University, Beijing, China [3]School of Computer Science and Technology, Beijing Institute of Technology, Beijing [4]School of Computer Science and Technology, Harbin Institute of Technology, Shenzhen, China [5]Baidu Inc., Beijing, China. Correspondence to: Zeming Liu <zmliu@buaa.edu.cn>.

*Proceedings of the 43rd International Conference on Machine Learning*, Seoul, South Korea. PMLR 306, 2026. Copyright 2026 by the author(s).

[1]Dataset and codes are publicly available at https://github.com/BUAA-IRIP-LLM/DocOS.

## 1. Introduction

Recent advances in Multimodal Large Language Models (MLLMs) have driven rapid progress in Graphical User Interface (GUI) agents, enabling automated interaction across a wide range of applications (Wang et al., 2024b; Nguyen et al., 2025; Yin et al., 2024; Lu et al., 2025; Liu et al., 2025), including web navigation (Yao et al., 2022; Deng et al., 2023; Zhou et al., 2024), desktop environments (Xie et al., 2024), and mobile devices (Zhang et al., 2023; Wang et al., 2024a). These developments mark an important step toward general agents capable of assisting humans with diverse real-world tasks.

Despite this progress, a fundamental limitation persists: existing GUI agents predominantly rely on *static parametric knowledge* acquired during pre-training or instruction tuning (Xu et al., 2025a; Liu et al., 2024; Zhou et al., 2025). While sufficient for common interactions, such knowledge is inherently incomplete and quickly becomes brittle when agents are confronted with long-tailed, application-specific tasks. These tasks often require explicit procedural instructions, precise option semantics, or rarely used functionalities that are absent from the model's fixed parameters. As illustrated in Figure 1, for a specific instruction like "Create a run configuration from a python template," the agent lacks the inherent knowledge to navigate the PyCharm interface directly. Instead, it must first navigate the open web to retrieve the official "Run/Debug Configurations" guide, utilizing this external document to ground its subsequent actions in the software. This behavior increases execution failures and hallucinations, fundamentally undermining the reliability and deployability of GUI agents in dynamic environments.

To mitigate this limitation, we introduce **Proactive Document-Guided Action**, a novel paradigm that mirrors human problem-solving by enabling agents to autonomously search the open web for documentation to resolve long-tailed tasks. Under this paradigm, agents are no longer restricted to static internal knowledge. Instead, they proactively navigate browsers to locate and comprehend documents in real time. This paradigm decouples an agent's reasoning capability from the storage of vast, domain-specific knowledge. By dynamically acquiring explicit procedural guidance on the fly, agents can faithfully ground their ac-

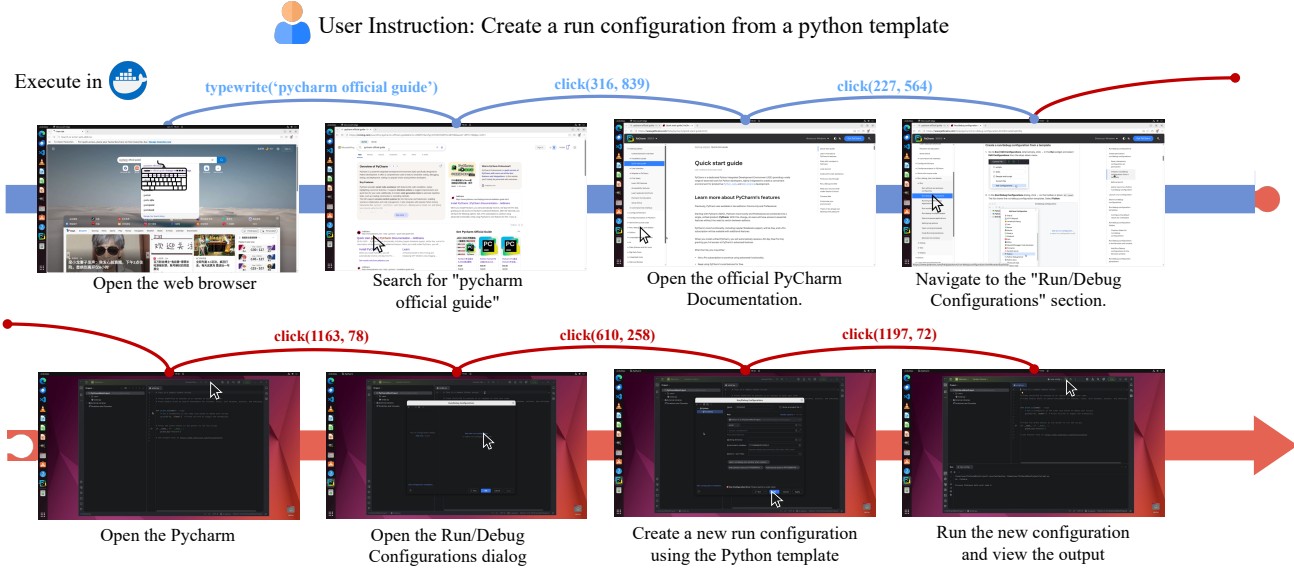

*Figure 1.* An example of a Proactive Document-Guided Action task. The workflow is divided into two distinct phases: Proactive Knowledge Retrieval (top row) and Document-Grounded Execution (bottom row).

tions in up-to-date documentation, enabling adaptation to long-tailed and evolving tasks without extensive retraining.

Guided by this paradigm, we propose **DocOS**, a benchmark designed to assess document-guided problem solving in fully interactive environments. Specifically, we curate a suite of high-difficulty instructions targeting long-tailed functionalities that are typically unsolvable without external documents. Crucially, we establish a dynamic execution environment (i.e., Docker) (Xie et al., 2024) where agents are not passively fed information and more details are in Appendix A. Instead, consistent with our paradigm, they are required to *autonomously navigate a web browser, locate relevant online documentation, comprehend procedural instructions, and faithfully ground them into executable GUI actions*. This setup moves beyond static, information-provided evaluations by explicitly assessing an agent's end-to-end capability to autonomously search relevant web documentation and ground the acquired procedural documents into precise GUI actions.

To systematically evaluate current GUI agents under this paradigm, we conduct extensive experiments on DocOS. Specifically, we design two distinct task settings to decouple and assess the core capabilities **Proactively Document Search**, which evaluates the agent's ability to locate documents via web browsing, and **Document-Grounded Execution**, which tests its capacity to implement instructions given the documentation. Results reveal that GUI agents are strictly constrained by dual bottlenecks: agents struggle to reliably locate relevant information during the proactive search phase, and even when provided with oracel document in the execution phase, they frequently fail to faithfully

ground retrieved instructions into precise actions. These findings highlight critical gaps in both document retrieval and document-conditioned execution, limiting end-to-end reliability in real-world deployment.

Overall, our contributions are summarized as follows:

- To the best of our knowledge, we are the first to propose **Proactive Document-Guided Action** for GUI agents, a novel paradigm that mirrors human problem-solving by enabling agents to autonomously search the open web for documents and apply them to solve long-tailed tasks.

- Guided by this paradigm, we introduce **DocOS**, a benchmark designed to evaluate GUI agents to autonomously navigate a web browser, locate relevant online documentation, comprehend procedural instructions, and accurately translate them into actions.

- We perform a systematic evaluation under two complementary settings: **Proactive Document Search** and **Document-Grounded Execution**. Our results show that the evaluated agents are fundamentally limited by **dual bottlenecks**: inaccurate information localization during document search and unfaithful grounding of instructions into actions, underscoring a critical gap toward robust, self-guided deployment.

## 2. Related Work

### 2.1. GUI Agents

Recent advances in MLLMs have enabled agents to perceive, reason about, and interact with GUI across diverse

*Table 1.* Comparison of virtual agent benchmarks. The columns indicate whether a benchmark supports proactive document search, document-grounded execution, interaction with dynamic web environments (i.e., non-static web pages), complex multi-step tasks, and fine-grained subtask-level evaluation.

| | Proactively Document Search | Document-Grounded Execution | Dynamic Web Environment | Complex Task | Subtask Evaluation |
|---|---|---|---|---|---|
| OmniACT (Kapoor et al., 2024) | ✗ | ✗ | ✗ | ✗ | ✗ |
| ScreenSpot-Pro (Li et al., 2025) | ✗ | ✗ | ✗ | ✗ | ✗ |
| WebArena (Zhou et al., 2024) | ✗ | ✗ | ✗ | ✗ | ✗ |
| VisualWebArena (Koh et al., 2024) | ✗ | ✗ | ✗ | ✗ | ✗ |
| OSWorld (Xie et al., 2024) | ✗ | ✗ | ✗ | ✗ | ✗ |
| CRAB (Xu et al., 2025b) | ✗ | ✗ | ✗ | ✗ | ✓ |
| VideoGUI (Lin et al., 2024) | ✗ | ✗ | ✗ | ✓ | ✓ |
| GUI-World (Chen et al., 2024) | ✗ | ✗ | ✗ | ✓ | ✗ |
| AitW (Rawles et al., 2023) | ✗ | ✗ | ✗ | ✓ | ✗ |
| Mind2Web (Deng et al., 2023) | ✗ | ✗ | ✗ | ✓ | ✗ |
| Spider2-V (Cao et al., 2024) | ✗ | ✗ | ✓ | ✓ | ✗ |
| WebCanvas (Pan et al., 2024) | ✗ | ✗ | ✓ | ✓ | ✓ |
| OmniBench (Bu et al., 2025) | ✗ | ✗ | ✓ | ✓ | ✓ |
| **DocOS (Ours)** | ✓ | ✓ | ✓ | ✓ | ✓ |

digital environments, rather than being limited to HTML or text based interfaces. Building on these foundations, a growing num of works focuses on specialized GUI agents designed for interactive environments, such as OS-Atlas (Wu et al., 2024b), DeepMiner-Mano (Fu et al., 2025), Open-CUA (Wang et al., 2025), CogAgent (Hong et al., 2024), OS-Copilot (Wu et al., 2024a) and Mobile-Agent (Ye et al., 2025) leveraging high-resolution visual perception to locate and interact with UI elements directly. However, most existing GUI agents implicitly rely on static parametric knowledge, and encounter great challenge when require konwledge beyond the model's training distribution. To release this issue, several approaches (Xie et al., 2025; Zhang et al., 2025) introduce retrieval-augmented mechanisms, allowing agents to access additional information at inference time. But these methods typically assume that relevant documents are pre-collected, and do not capture the challenge of autonomous knowledge acquisition in open environments.

## 2.2. Benchmarks for GUI Agent

Current GUI benchmarks in Table 1 proposed for GUI Agents primarily focus on evaluating their ability to accomplish GUI-based tasks under different interaction settings. Recent benchmarks also evaluate broader autonomous agent capabilities, including proactivate conversation (Liu et al., 2020; Shi et al., 2023; Liu et al., 2021; 2022), translation (Li et al., 2020; Chen et al., 2025), and tool-use (Liang et al., 2025; Li et al., 2026), but overlook proactive document-guided GUI interaction. Static benchmarks like Mind2Web (Gou et al., 2025), OmniACT (Kapoor et al., 2024), and ScreenSpot-Pro (Jurmu et al., 2008) are constructed from offline data and assess the agent's ability to predict actions or trajectories without real-time environment feedback, which do not capture the challenges of interac-

tive execution. In contrast, dynamic benchmarks including OSWorld (Xie et al., 2024), WebArena (Zhou et al., 2024) provide realistic, fully interactive environments that allow agents to execute actions, observe real-time feedback, and adapt their strategies accordingly. However, they typically assume that agents already possess the knowledge required to complete tasks and remain a gap of self-guided deployment. While Spider2-V (Cao et al., 2024), WebCanvas (Pan et al., 2024), and OmniBench (Bu et al., 2025) build dynamic online environment to evaluate web navigation tasks, they are limited to information retrieval and do not assess document-grounded execution. To the best of our knowledge, no existing benchmark systematically evaluates the end-to-end capability of GUI agents to actively search for updating frequently online documentation and ground retrieved procedural knowledge into executable GUI actions within a unified interactive environment. To alleviate this problem, we introduce DocOS, which evaluating GUI agents in Autonomous Document Search and Document-Grounded Execution.

## 3. DocOS

To evaluate the ability of GUI agents to retrieval in dynamic web environment and execute following the retrieval results, we construct DocOS that consists 817 high-quality desktop tasks across 20 applications to evaluate the ability of agents to search, retrieve, understand external online resources and execute complex, multi-step actions in realistic desktop environments.

## 3.1. Task Formulaton

We formalize the interaction between the GUI agent and the digital environment as a Partially Observable Markov

Decision Process (POMDP), defined by the tuple $\mathcal{M} = \langle \mathcal{S}, \mathcal{A}, \mathcal{O}, \mathcal{T}, \mathcal{I}, \mathcal{G}, \mathcal{V} \rangle$. Here, $\mathcal{S}$ represents the latent states of the operating system and applications, $\mathcal{A}$ denotes the space of permissible actions (e.g., clicking, typing, scrolling), and $\mathcal{O}$ is the observation space consisting of visual screenshots and accessibility trees. $\mathcal{T} : \mathcal{S} \times \mathcal{A} \rightarrow \mathcal{S}$ represents the transition dynamics of the environment, $\mathcal{I}$ is the natural language user instruction, $\mathcal{G}$ is the goal conditions which defining the expected state for task completion, and $\mathcal{V} : \mathcal{S} \times \mathcal{G} \rightarrow 0, 1$ represents a verifier that determines whether the current state satisfies the goal.

At each time step $t$, the agent receives an observation $o_t \in \mathcal{O}$ derived from the current state $s_t$ and executes an action $a_t \in \mathcal{A}$ based on its policy $\pi$. The environment then transitions to a new state $s_{t+1}$ according to $\mathcal{T}$.

In the context of the **Proactively Document-Guided Action** task, the agent cannot rely solely on internal parametric knowledge to solve $\mathcal{I}$ due to the long-tailed or domain-specific nature of the task. Instead, the process is decomposed into two distinct sequential phases: *Proactive Knowledge Retrieval* and *Document-Grounded Execution*.

**Proactive Knowledge Retrieval**   In this initial phase, the agent's objective is not to immediately fulfill the user instruction $\mathcal{I}$, but to actively acquire external information required to solve it. The agent interacts with a web browser environment to search for a relevant manual or documentation. The policy for this phase, denoted as $\pi_{\mathrm{retr}}$, generates actions to navigate search engines and help centers:

$$a_t \sim \pi_{\mathrm{retr}}(a_t \mid o_t, h_{t-1}, \mathcal{I}), \quad \text{for } 0 \leq t < T_{\mathrm{retr}} \quad (1)$$

where $h_{t-1}$ represents the interaction history. This phase concludes at time step $T_{\mathrm{retr}}$ when the agent identifies and retrieves a target document $\mathcal{D}$, which contains the necessary procedural knowledge or semantic explanations.

**Document-Grounded Execution**   Upon retrieving the document $\mathcal{D}$, the agent transitions to the execution phase. The goal shifts to completing the original user instruction $\mathcal{I}$ by grounding the actions in the retrieved context $\mathcal{D}$. The policy $\pi_{\mathrm{exec}}$ operates as follows:

$$a_t \sim \pi_{\mathrm{exec}}(a_t \mid o_t, h_{t-1}, \mathcal{I}, \mathcal{D}), \quad \text{for } t \geq T_{\mathrm{retr}} \quad (2)$$

Unlike standard GUI agents that depend solely on $\mathcal{I}$, our formulation explicitly conditions the action generation on the unstructured knowledge $\mathcal{D}$. The task is considered successful if and only if the final state $s_{End}$ satisfies the goal condition specified by $\mathcal{I}$, verifying that the agent has correctly interpreted and applied the external knowledge.

### 3.2. Data Collection

To construct a scalable and semantically meaningful task set for GUI agents, the data collection pipeline consists of three

stages as shown in Figure 2, including task construction, document collection, and task filtering. This pipeline leverages authoritative documentation as the grounding source and combines task construction with careful filtering.

**Task Construction**   We manually construct task instructions along with their corresponding prerequisity to ensure high quality and realism. The tasks are designed to be realistic and goal-oriented, closely reflecting how users would interact with the application in practice. Following current works (Cao et al., 2024; Koh et al., 2024), we categorize tasks into three difficulty levels based on the number of required execution steps: Easy tasks can be completed within four steps or fewer, typically involving direct and atomic interactions; Middle tasks require five to seven steps, often involving multiple interface components or conditional actions; Hard tasks consist of eight or more steps, representing complex workflows that require sustained planning and multi-stage interaction.

**Document Collection**   We collect authoritative official documentation from a diverse set of real-world applications. These documents are sourced from publicly available official websites, which provide reliable and comprehensive descriptions of application functionalities, user workflows, and interface behaviors. To ensure scalability and consistency across applications, we design an automated crawling and preprocessing script that automatically retrieves and cleans the raw web pages for each application. The script removes non-informative elements such as navigation bars, advertisements, and redundant boilerplate content, while preserving semantically meaningful components including functional descriptions, step-by-step instructions, and UI-related explanations. The resulting cleaned documents serve as structured and reliable reference for Proactive knowledge retrieval.

**Task Filtering**   The tasks suffer from issues such as semantic ambiguity, and infeasible execution. To alleviate these problems, we introduce the task filtering stage that removes the low-quality tasks. Specifically, we assess whether each task is (1) semantically consistent with the source documentation, (2) clearly specified and unambiguous, and (3) realistically executable in a real application environment. As a result, we select 817 high-quality samples from thoustand of candidates.

### 3.3. Quality Control and Validation

We adopt a multi-stage quality control pipeline that combies cross-verification, validation of external resources to improve task executability, clarity and evaluation stability. Annotators execute the task following the provided instructions and external instructional resources within the target

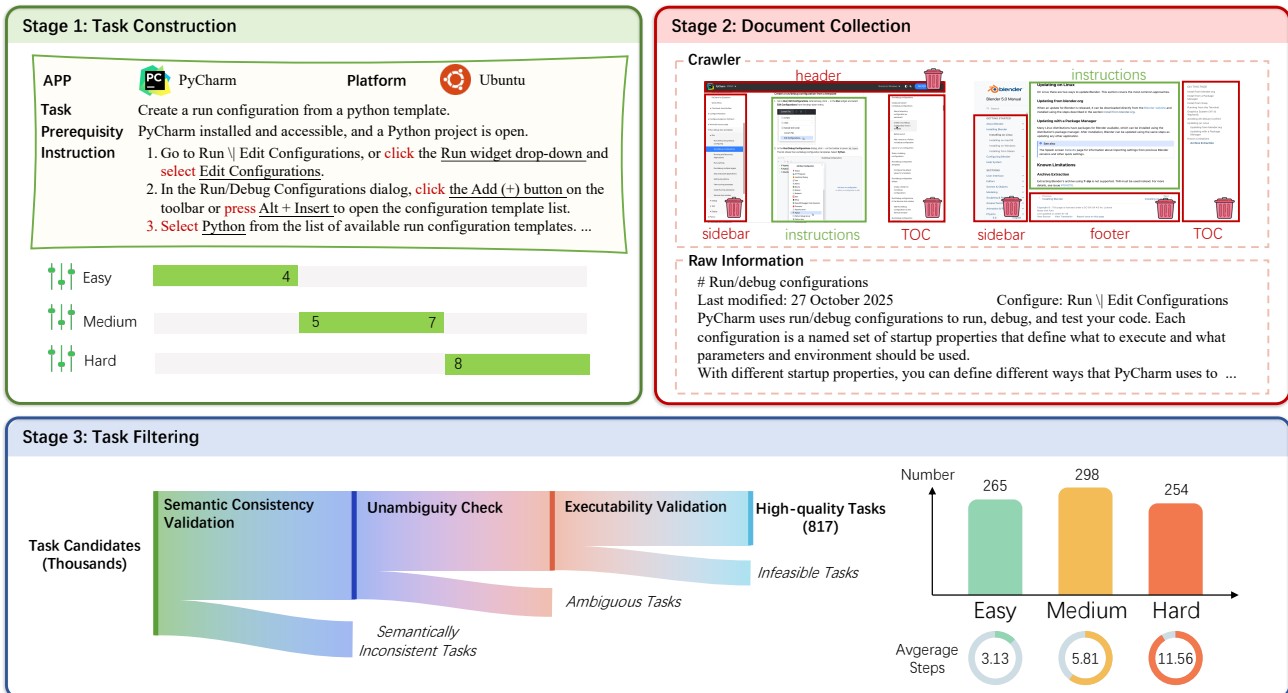

*Figure 2.* The data construction pipeline of **DocOS** consists of 3 stages. **Stage 1: Task Construction** involves defining task instructions, prerequisites, and difficulty levels (Easy, Medium, Hard) based on execution steps. **Stage 2: Document Collection** utilizes an automated crawler to retrieve and parse official documentation, extracting structured raw information (e.g., instructions, headers). **Stage 3: Task Filtering** implements a rigorous quality control funnel—validating semantic consistency, unambiguity, and executability—to select 817 high-quality tasks from thousands of candidates.

software environment, and report whether the task can be completed, as well as any ambiguities, or failures encountered during execution. To verify external resource validity stability, and relevance to the task objective, we perform both automated accessibility checks and manual inspection. Resources that are inaccessible, unstable, or weakly aligned with the task goal are removed to prevent failures unrelated to agent performance. This preprocessing step is only used to construct the ground-truth document; the agent evaluation environment remains fully dynamic and unfiltered. Following previous work (Xie et al., 2024), We sample approximately 20% (160 tasks) of the data to conduct human evaluations for data quality and find that 92.50% (148 tasks) of the data satisfies the requirements of semantic consistency, unambiguous instructions, and executability.

### 3.4. Data Statistics

As illustrated in Table 2, DocOS comprises 817 task instances spanning 20 applications, which are divide into three difficulty levels: Easy, Medium, and Hard. The number of tasks is relatively balanced among the three levels, with 265 easy tasks, 298 medium tasks, and 254 hard tasks. More statistic details and data examples in DocOS can be found in Appendix B.

*Table 2.* Statistic of DocOS.

| Statistic | Level | | | Total |
|---|---|---|---|---|
| | Easy | Medium | Hard | |
| # APP | 15 | 17 | 19 | 20 |
| # Tasks | 265 | 298 | 254 | 817 |
| # Avg. Steps | 3.13 | 5.81 | 11.56 | 6.72 |
| # Avg. Prompt Len. | 886.54 | 1286.69 | 2204.13 | 1442.12 |

## 4. Experiments

### 4.1. Experiment Setting

**Baselines** Following prior work (Bu et al., 2025), we conduct a comprehensive experiment on **DocOS** using Qwen3-VL-8B (Bai et al., 2025), UI-TARS-1.5-7B (Qin et al., 2025), MAI-UI-8B (Zhou et al., 2025), GELab-Zero-4B-preview (Yan et al., 2025), Aguvis-7B (Xu et al.), and GUI-Owl-7B (Ye et al., 2025).

**Metrics** To comprehensively evaluate the capabilities of DocOS, we divide the evaluation process into two distinct phases: (1) **Documentation Search**, where the agent must navigate to the correct target page to retrieve necessary information, and (2) **Task Execution**, wherein the agent must accomplish the specific user objective based on the

*Table 3.* **Main results on Easy, Medium, and Hard subsets.**

| Model | Easy | | | Medium | | | Hard | | | Avg. | | |
|---|---|---|---|---|---|---|---|---|---|---|---|---|
| | TUI↑ | HPP↑ | TCR↑ | TUI↑ | HPP↑ | TCR↑ | TUI↑ | HPP↑ | TCR↑ | TUI↑ | HPP↑ | TCR↑ |
| GELab-Zero-4B-preview | 5.88 | 51.47 | 5.17 | 0.00 | 47.83 | 6.63 | 5.88 | 62.25 | 2.12 | 3.23 | 52.78 | 4.36 |
| UI-TARS-1.5-7B | 4.26 | 56.12 | 30.86 | 1.91 | 54.41 | 17.32 | 1.20 | 56.08 | 13.08 | 2.48 | 55.48 | 17.31 |
| MAI-UI-8B | 3.00 | 47.47 | 20.99 | 2.62 | 45.66 | 7.82 | 3.10 | 42.83 | 10.00 | 2.89 | 45.34 | 10.96 |
| Aguvis-7B | 0.00 | 0.00 | 5.97 | 0.00 | 0.00 | 1.23 | 0.00 | 0.00 | 1.29 | 0.00 | 0.00 | 1.95 |
| GUI-Owl-7B | 0.00 | 13.97 | 9.88 | 0.00 | 16.79 | 6.15 | 5.00 | 21.67 | 3.85 | 1.54 | 17.33 | 5.58 |
| Qwen3-VL-8B | 44.53 | 51.07 | 7.41 | 39.60 | 55.89 | 3.35 | 42.91 | 58.82 | 3.85 | 42.23 | 55.21 | 4.23 |

retrieved context. Accordingly, we design specific metrics for each phase to ensure a granular assessment.

**Phase 1: Documentation Search Metrics** This phase assesses the agent's ability to perceive visual cues and execute correct hierarchical navigation sequences. Since the agent operates on raw pixels, we utilize URL-based metrics as an objective proxy for navigation precision.

**Target URL Inclusion (TUI)** TUI measures whether the agent successfully locates the target documentation. **Justification:** In dynamic web environments, servers often append variable parameters (e.g., session IDs, tracking tokens) to URLs, making exact string matching overly rigid. Therefore, we adopt an inclusion-based criterion: a search is considered successful if the ground truth URL is a substring of the agent's final retrieved URL.

$$TUI = \frac{1}{N} \sum_{i=1}^{N} \mathbb{I}(U_{gt}^{(i)} \subseteq U_{retrieved}^{(i)}) \tag{3}$$

where $N$ is the total number of tasks, $U_{gt}$ and $U_{retrieved}$ denote the ground truth and retrieved URLs respectively, and $\mathbb{I}(\cdot)$ is the indicator function.

**Hierarchical Path Progress (HPP)** HPP evaluates the "depth" of the correct navigation path achieved by the agent. We treat a URL as a sequence of directory segments $\{s_1, s_2, ..., s_k\}$ delimited by forward slashes ('/'). HPP is the ratio of the matching prefix length to the total GT path length:

$$HPP = \frac{1}{N} \sum_{i=1}^{N} \frac{|\text{LCP}(S_{gt}^{(i)}, S_{retrieved}^{(i)})|}{|S_{gt}^{(i)}|} \tag{4}$$

where $\text{LCP}(\cdot)$ denotes the Longest Common Prefix. **Justification:** Binary success metrics (like TUI) fail to capture partial progress. HPP allows us to distinguish between agents that are "completely lost" and those that navigate correctly through the site hierarchy but fail at the final leaf node, providing deeper insight into the agent's reasoning logic.

**Phase 2: Task Execution Metrics** This phase evaluates the end-to-end utility of the agent, focusing on whether the user's ultimate intent was satisfied.

**Task Completion Rate (TCR)** TCR quantifies the overall success rate of the system. It is defined as the ratio of successfully completed tasks to the total number of evaluation tasks:

$$TCR = \frac{N_{success}}{N_{total}} \tag{5}$$

where $N_{total}$ is the total number of tasks and $N_{success}$ is the count of tasks where the final state meets the acceptance criteria. While navigation metrics (TUI/HPP) measure intermediate process correctness, TCR serves as the decisive metric for the agent's practical utility in real-world scenarios.

**Implementation Details** The experiments are conducted on a server equipped with NVIDIA RTX 4090 GPUs. Multi-GPU parallelism is employed to accelerate inference. During the experiments, the web navigation process for the official documentation is executed for 15 steps, which is used to acquire task-relevant prior knowledge and construct candidate action strategies. Then, the task execution phase is carried out for 50 steps to validate and evaluate the effectiveness of the final strategy. All experiments are repeated under same hardware configurations to ensure fairness and reproducibility. The prompt used in the experiments is shown in Appendix F.

### 4.2. Main Results

The main experimental results of different GUI agents on the DocOS are shown in Table 3. The evaluation is conductedd across three difficulty levels (Easy, Medium, and Hard).

**All evaluated agents exhibit limited end-to-end task completion performance on DocOS.** Most agents achieve limited TCR even on the Easy subset, and performance consistently degrades as task difficulty increases. These results indicate that DocOS presents a challenging evaluation setting that effectively exposes the limitations of existiing

agents on long-tailed, application-specific, and multi-step GUI workflows. Meanwhile, several models maintain relatively stable TUI and HPP across difficulty levels, indicating that they are able to issue queries and make partial progress toward relevant documentation or target paths. However, such navigation-level progress frequently fails to translate into successful task completion. The experiments results suggest that the reliability of subsequent document-guided execution remain limited under the DocOS setting.

**While the agents are generally able to access official documentation websites, they struggle to precisely localize task-relevant information within these websites.** Across most agents and levels, the stable HPP scores suggest that agents are often able to reach higher-level sections of the target paths. However, the consistently low TCR implies that these agents frequently fail to identify the specific pages, sections within the documentations that directly correspond to the required operation. As a result, limitaions in navigating within the documentation websites and locating the exact information that provides actionable guidance emerge as a major factor associated with low task completion performance.

## 5. Analysis

In this section, we conduct a comprehensive analysis to answer the following research questions **RQ1:** *What is the importance of documents in GUI agent performance?* (§5.1) **RQ2:** *What are the bottlenecks in the GUI agents' capabilities?* (§5.2) **RQ3:** *How do document length and step size affect task completion?* (§5.3) **RQ4**: *How does the GUI agent behave in case studies?* (§5.4)

### 5.1. RQ1: Importance of Documents

*Table 4.* Performance comparison of various GUI agents with and without access to external documents. The last column indicates the relative performance change ($\Delta_{rel}$).

| Models | w/o Document | w/ Document | $\Delta_{rel}^{\%}$ |
|---|---|---|---|
| GELab-Zero-4B-preview | 4.25 | 4.36(+0.11) | ↑ 2.59% |
| UI-TARS-1.5-7B | 16.70 | 17.31(+0.61) | ↑ 3.65% |
| MAI-UI-8B | 10.18 | 10.96(+0.78) | ↑ 7.66% |
| Aguvis-7B | 1.91 | 1.95(+0.04) | ↑ 2.09% |
| GUI-Owl-7B | 5.27 | 5.58(+0.31) | ↑ 5.88% |
| Qwen3-VL-8B | 4.01 | 4.23(+0.22) | ↑ 5.49% |

To verify the effectiveness of the Document-Guided Action task, we investigate the impact of introducing document guidance on the performance of GUI agents in task execution. Specifically, we have the agents skip the *Autonomous Knowledge Retrieval* phase as described in Section 3.1 and directly rely on the model's parameterized knowledge to perform the tasks.

The results of this comparative analysis are presented in Table 4. As observed, integrating external documents generally enhances the performance of most GUI agents, validating the necessity of the Document-Guided Action task. In detail, **MAI-UI-8B** demonstrates the most significant benefit from document guidance, achieving a relative improvement of ↑ 7.67%, followed by **GUI-Owl-7B** with a ↑ 5.88%, and **Qwen3-VL-8B** with a ↑ 5.49% increase. Notably, even the strongest baseline, **UI-TARS-1.5-7B**, which possesses robust parameterized knowledge, sees a performance gain of ↑ 3.67%, indicating that external documentation effectively supplements internal knowledge gaps even for advanced models. However, the impact varies across model capacities. **Aguvis-7B** shows less performance increase, suggesting a potential bottleneck in its ability to utilize long-context information. The smaller model, **GELab-Zero-4B-preview**, experiences a performance increase of ↑ 2.59%. This situation likely stems from the model's limited context window or reasoning capacity, where the additional document content introduces noise or distraction rather than guidance. Overall, for models with sufficient comprehension capabilities, the introduction of documents serves as a critical bridge for accurate task execution.

### 5.2. RQ2: Bottleneck in GUI Agents

*Table 5.* Performance comparison of different models using the full pipeline (w/ Document) versus using ground-truth instructions (Oracle Document).

| Models | w/ Document | Oracle Document | $\Delta_{rel}^{\%}$ |
|---|---|---|---|
| GELab-Zero-4B-preview | 4.36 | 4.83(+0.47) | ↑ 10.78% |
| UI-TARS-1.5-7B | 17.31 | 17.98(+0.67) | ↑ 3.87% |
| MAI-UI-8B | 10.96 | 12.04(+1.08) | ↑ 9.85% |
| Aguvis-7B | 1.95 | 2.1(+0.15) | ↑ 7.69% |
| GUI-Owl-7B | 5.58 | 5.81(+0.23) | ↑ 4.12% |
| Qwen3-VL-8B | 4.23 | 4.44(0.21) | ↑ 4.96% |

In this section, we delve into the bottlenecks of the agents in the Document-Guided Action task. By eliminating the need for document retrieval, we directly evaluate the agent's ability to execute tasks when provided with reliable, pre-provided instructions (i.e., an oracle document).

Table 5 reports the results of providing agents with Oracle Documents (ground-truth instructions) compared to the standard setting. The results reveal a divergent trend that highlights distinct bottlenecks across different models. On one hand, **MAI-UI-8B** and **GELab-Zero-4B-preview** exhibit substantial performance gains, with relative improvements of ↑ 9.85% and ↑ 10.78%, respectively. This drastic increase suggests that the primary bottleneck for these models in the standard setting is the *accuracy of information acquisition*. When the noise of retrieval is removed and correct instructions are provided, these models demonstrate strong grounding and execution capabilities, proving they

are capable "followers" but weaker "planners."

On the other hand, unexpectedly, strong baselines like **UI-TARS-1.5-7B** (↑ 3.87%) and **Qwen3-VL-8B** (↑ 4.96%) experience a less significant performance increase when provided with Oracle documents. This counter-intuitive result points to a *context processing bottleneck*. Despite having the correct information, these models likely struggle to filter relevant actions from the detailed Oracle descriptions, or the lengthy context acts as a distraction that interferes with their inherent reasoning capabilities. This indicates that for some models, simply feeding "correct" knowledge is insufficient if the model lacks the robustness to handle long-context instructions effectively.

### 5.3. RQ3: Impact of the Document Length and Step Size

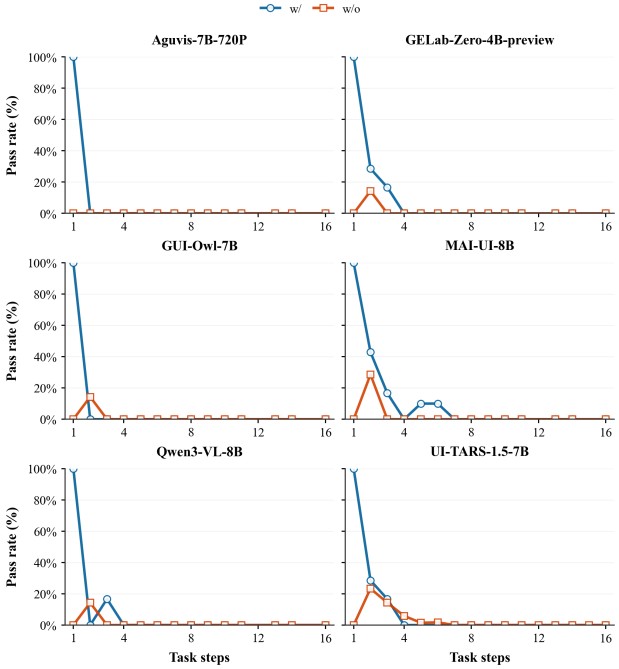

*Figure 3.* Pass rates of six different GUI agents across varying task steps under two settings: with (w/) and without (w/o) provided documents. The x-axis represents the number of steps required to complete the task, and the y-axis represents the pass rate.

In this section, we analyze how varying the length and step size of the documents provided to the GUI agent influences its task completion performance. As shown in Figure 3, we observe a significant negative correlation between the number of task steps and the agent's success rate across all six evaluated models.

Specifically, in the setting with documents (**w/**), most models achieve a near 100% pass rate for single-step tasks. However, the performance degrades sharply as the task horizon extends. For tasks requiring more than 4 steps, the pass rate drops to nearly 0% for almost all models, indicating that

current agents struggle significantly with long-horizon planning and context retention, even when auxiliary documents are provided.

In contrast, the setting without documents (**w/o**) shows consistently lower performance. Notably, for 1-step tasks, the w/o setting often yields a near 0% pass rate, suggesting that without external guidance, agents fail to initiate the correct action for specific UI tasks.

### 5.4. RQ4: Error Analysis

To understand the limitation of current GUI agents in the document-guided action setting, We randomly sampled a subset of failed cases, conduct a detailed error analysis on them, and illustrate the primary error types during two stages. Detailed cases are in Appendix C.

During the document search and navigation stage, the errors arised can be categorized into several recurring patterns. **(1) Imprecise Localization** GUI agent fail to precisely localize task-relevant content within documentation. This is the most prominent and impactful failure mode in the failed cases. While agents are often able to reach the official documentation website and navigate to high-level pages, the frequently remain on loosely relevant section. **(2) Non-official Reference** In some failure cases, GUI agents retrieve information from third-party blogs, or tutorials instead of authoritative documentation. These sources often contain incomplete instructions, leading agents to perform incorrect actions. **(3) Execute before Retrieval** GUI agent bypasses documentation search and directly attempting task execution. Some failed cases ignore the guidance and proceed to execute the task immediately. Such behavior results in incorrect action sequences based on their parametric knowledge.

For GUI agents executing tasks based on retrieved search results, we find that the primary causes of failure are as follows. **(1) Action Grounding Failure** There is difficulty for GUI agent to translate instructions into concrete verifiable GUI actions. Even when agents access relevant interfaces, GUI agent stop at navigation without performing the specific action such as parameter adjustments. This reveals a gap between high-level instruction recognition and low-level action grounding. **(2) Context Misidentification** GUI Agent fail to identify the correct operational context. For example, it did not enter the required mode (e.g., Edit Mode) before attempting further actions. As a result, subsequent operations that depend on object mode were never triggered.

## 6. Conclusion

To improve the evaluation of GUI agents, we introduced Proactive Document-Guided Action, a document-driven paradigm that addressed the limitation of existing GUI agent evaluations by requiring agents to actively retrieve exter-

nal documentation when solving long-tailed, application-specific tasks. Within this paradigm, we proposed DocOS, a benchmark that evaluates GUI agents across the full pipeline of retrieval, understanding, and execution in realistic desktop environments. The experiment results show that GUI agents struggle to precisely localize task-relevant data and translate retrieved instructions into executable GUI actions, which provides valuable insights into the key bottlenecks that limit reliable, self-guided deployment of GUI agents. While these observations are contextualized within the specific architectures and evaluation benchmarks explored in our work, we provide a foundational understanding of the challenges that future GUI agents must overcome.

## Impact Statement

DocOS aims to evaluate GUI agents under the noval paradigm of proactive document-guided action, which can enhance the reliability of GUI agents to resolve long-tailed tasks. We hope DocOS provides researchers with a systematic benchmark to better understand the strengths and limitations of GUI agents, and we strongly encourage responsible and ethical research and deployment of GUI agent systems.

## Acknowledgments

Thanks for the insightful comments and feedback from the reviewers. This work was supported by the National Natural Science Foundation of China (No. 62406015). Jingjing Liu, Ziye Huang, and Zihao Cheng contributed equally to this work. Zeming Liu is the corresponding author.

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

# A. Details of DocOS Environment

## A.1. Infrastructure

Inspired by OSWorld (Xie et al., 2024), we construct an interactive desktop environment to support the evaluation of GUI agents. The environment is managed througy Docker for containerized deployment and leverages QEMU for system-level virtualization, enabling a fully controllable and flexible execution setting. Specifically, to reduce storage overhead, improve execution efficiency, and enable flexible composition of software environments, we construct lightweight and modular filesystem images, each encapsulating a single target application along with its required dependencies.

## A.2. Web Search

DocOS environment provides controlled Internet access to support instruction-driven online information retrieval. Network connectivity is enabled, and all web browsing and search activities are conducted exclusively through the Microsoft Edge browser, ensuring a realistic and consistent user interaction paradigm. During the search process, agents are required to formulate queries in a human-like manner, using natural-language keywords rather than structured search APIs or backend retrieval interfaces. In addition, the environment supports persistent recording of browsing states. Opened web pages are saved and screenshots are captured during task execution, providing visual records for result verification, behavior analysis, and reproducibility of the evaluation process.

## A.3. Observation Space

The observation space is defined by screen captures in DocOS, enabling the evaluation of GUI agents based on visual perception. Screen captures provide a complete visual snapshot of the desktop environment, including window layouts, widget appearances, textual content, and graphical elements, supplying all necessary visual cues for interface understanding and action decision-making.

## A.4. Action Space

*Table 6.* **Detailed Specification of the DocOS Action Space.** This table enumerates all atomic actions available to the agent. For each action type, we specify the required and optional (*opt*) parameters, their data types, valid ranges (based on a $1920 \times 1080$ screen resolution), and a functional description.

| Action Type | Parameters | Description |
|---|---|---|
| MOVE_TO | **x**: float $\in [0, 1920]$ 
 **y**: float $\in [0, 1080]$ | Move the cursor to the specified position. |
| CLICK | **button**: str $\in$ {left, right, middle} (opt) 
 **x**: float $\in [0, 1920]$ (opt) 
 **y**: float $\in [0, 1080]$ (opt) 
 **num_clicks**: int $\in$ {1, 2, 3} (opt) | Click the specified button (default: left). Click at current position if x, y are not specified. |
| MOUSE_DOWN | **button**: str $\in$ {left, right, middle} (opt) | Press the specified button (default: left) without releasing. |
| MOUSE_UP | **button**: str $\in$ {left, right, middle} (opt) | Release the specified button (default: left). |
| RIGHT_CLICK | **x**: float $\in [0, 1920]$ (opt) 
 **y**: float $\in [0, 1080]$ (opt) | Right click at the current position if x and y are not specified, otherwise at the specified position. |
| DOUBLE_CLICK | **x**: float $\in [0, 1920]$ (opt) 
 **y**: float $\in [0, 1080]$ (opt) | Double click at the current position if x and y are not specified, otherwise at the specified position. |

| Action Type | Parameters | Description |
|---|---|---|
| DRAG_TO | **x**: float $\in [0, 1920]$ 
 **y**: float $\in [0, 1080]$ | Drag the cursor to the specified position with the left button pressed. |
| SCROLL | **dx**: int 
 **dy**: int | Scroll the mouse wheel up or down. |
| TYPING | **text**: str | Type the specified text string. |
| PRESS | **key**: str $\in$ KEYS | Press the specified key and release it. |
| KEY_DOWN | **key**: str $\in$ KEYS | Press the specified key (hold down). |
| KEY_UP | **key**: str $\in$ KEYS | Release the specified key. |
| HOTKEY | **keys**: list[str] $\subseteq$ KEYS | Press the specified key combination simultaneously. |
| WAIT | *None* | Wait until the next action. |
| FAIL | *None* | Decide the task cannot be performed. |
| DONE | *None* | Decide the task is done. |

We define the action space of **DocOS** by drawing inspiration from the design paradigm of OSWorld (Xie et al., 2024). The space consists of **16 atomic actions** that encompass a comprehensive set of mouse and keyboard interactions, enabling the agent to operate the GUI like a human user. All low-level executions are implemented using the pyautogui[2] library to ensure robust and cross-platform compatibility. The detailed specifications of these actions, including their parameters and functional descriptions, are provided in Table 6.

## B. Details of DocOS Benchmark

### B.1. Data Statistic

As shown in table 7, we collected updating document from official websites of a diverse set of real-world software applications, spanning development environments, multimedia tools, collaboration platforms, and dataa analytics systems. For each application, we report the corresponding documentation domain, the average number of steps, and the average prompt length.

### B.2. Data Example

We provide detailed data examples in DocOS, including Figures 4-17. Each item is constructed with a task instruction, an expected document url, and several standard action steps. These examples cover a variety of document-grounded tasks with different levels of complexity.

## C. Error cases

This section presents qualitative cases of common error types made by GUI agents. Figures 18, 19, and 20 show the primary error types in the process of proactive document search. Figures 21 and 22 further illustrate the main error types encountered during the document-grounded execution.

---

[2]https://github.com/asweigart/pyautogui

*Table 7.* **Statistics of Tasks across Different Applications.** The table lists the application name, its corresponding domain, total task count, average steps to completion, and average prompt length.

| Application | Domain | Tasks | Avg. Steps | Avg. Prompt Len. |
|---|---|---|---|---|
| PyCharm | `https://www.jetbrains.com/help/pycharm/` | 257 | 6.92 | 1479.28 |
| Idea | `https://www.jetbrains.com/help/idea` | 163 | 6.45 | 1260.59 |
| Blender | `https://docs.blender.org/manual/en/latest/` | 125 | 5.16 | 975.52 |
| Postman | `https://learning.postman.com/docs/` | 94 | 7.23 | 1292.50 |
| Godot | `https://docs.godotengine.org/` | 41 | 7.29 | 3020.51 |
| Zotero | `https://www.zotero.org/support/` | 39 | 6.03 | 1125.85 |
| Zulip | `https://zulip.com/help/` | 20 | 7.75 | 819.55 |
| Element | `https://docs.element.io/latest/` | 13 | 8.31 | 965.69 |
| NetBeans | `https://netbeans.apache.org/tutorial/` | 13 | 9.75 | 3862.08 |
| VSCode | `https://code.visualstudio.com/docs/` | 12 | 8.42 | 3616.75 |
| Anki | `https://docs.ankiweb.net/` | 8 | 4.62 | 1413.50 |
| Odoo_19_0 | `https://www.odoo.com/documentation/19.0/` | 7 | 9.86 | 1502.29 |
| Grafana | `https://grafana.com/docs/grafana/` | 6 | 12.17 | 1172.33 |
| Emby | `https://emby.media/support/` | 4 | 6.50 | 1109.00 |
| Metabase | `https://www.metabase.com/docs/` | 4 | 7.00 | 1305.50 |
| Cryptomator | `https://docs.cryptomator.org/` | 3 | 3.33 | 897.67 |
| Notepad++ | `https://npp-user-manual.org/docs/` | 3 | 8.00 | 4206.33 |
| Signal | `https://support.signal.org/` | 3 | 8.33 | 759.67 |
| Geany | `https://www.geany.org/` | 1 | 10.00 | 1321.00 |
| VLC | `https://wiki.videolan.org/` | 1 | 8.00 | 1399.00 |

**App:** Anki

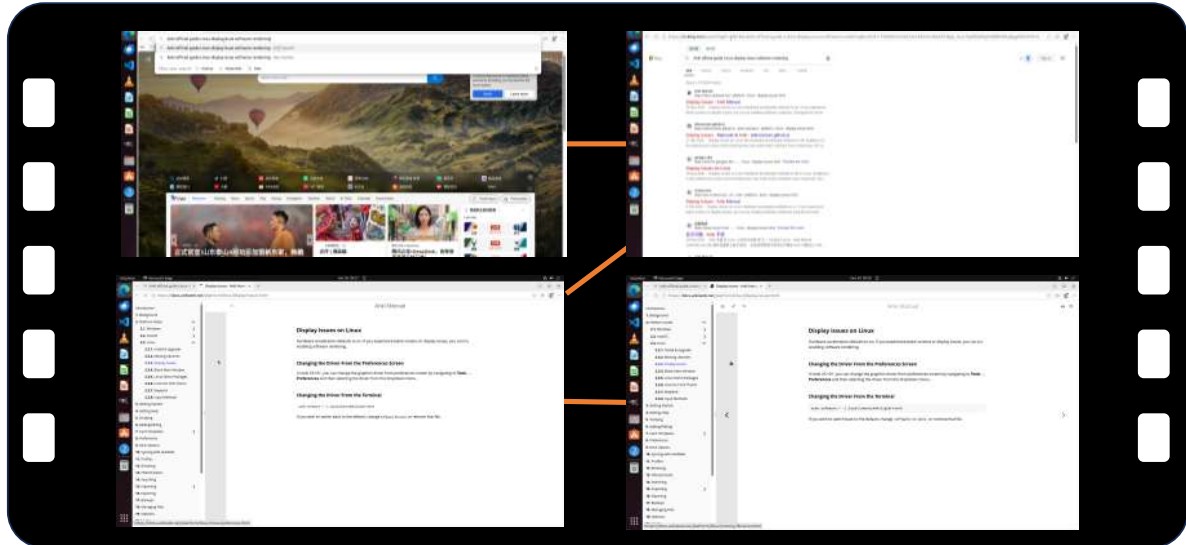

**Task Instruction**: Resolve Linux display issues in Anki by enabling software rendering, either via the GUI Preferences or a terminal command

**actual_url:**https://docs.ankiweb.net/platform/linux/display-issues.html
**expected_url:**https://docs.ankiweb.net/platform/linux/display-issues.html

**Steps:**
- Step_1: Open Anki, go to Tools → Preferences, and from the graphics driver dropdown select software rendering.
- Step_2: Alternatively, in a terminal, run: echo software > ~/.local/share/Anki2/gldriver6
- Step_3: To revert to the default driver, replace software with auto in the same file or remove that file

*Figure 4.* A sample of *Anki*.

**App:** Blender

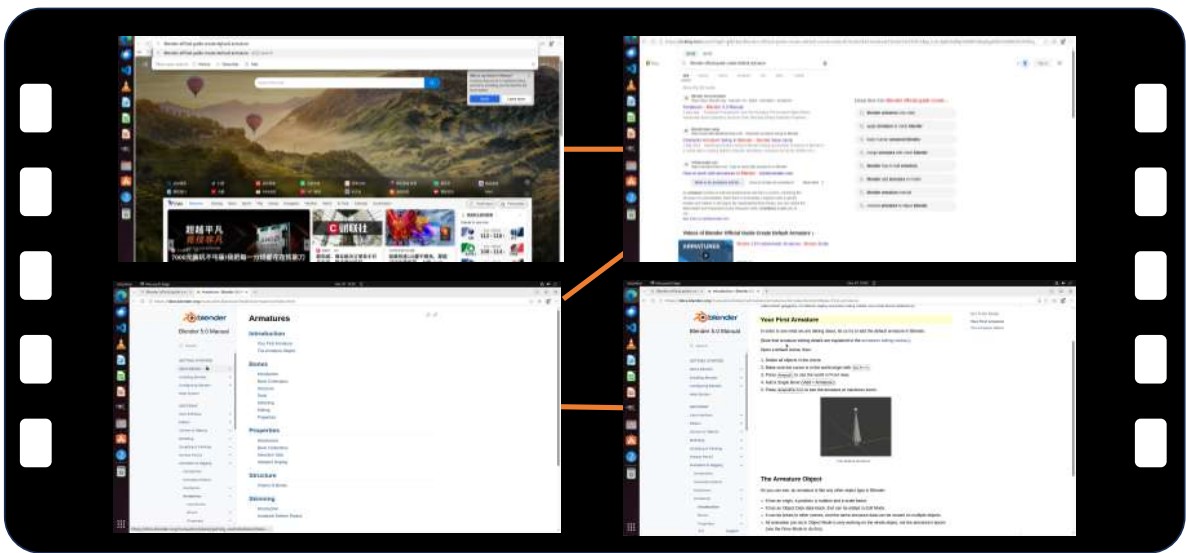

**Task Instruction:**Create a default armature in Blender by clearing the scene, centering the 3D cursor, switching to Front view, adding a single bone, and zooming in.

**actual_url:**https://docs.blender.org/manual/en/latest/animation/armatures/introduction.html#your-first-armature
**expected_url:**https://docs.blender.org/manual/en/latest/animation/armatures/introduction.html

**Steps:**
- Step_1: Delete all objects in the scene.
- Step_2: Ensure the 3D cursor is at the world origin using Shift-C.
- Step_3: Press Numpad1 to switch to Front view.
- Step_4: Add a Single Bone by choosing Add → Armature.
- Step_5: Press NumpadPeriod to maximize the zoom so the armature is fully visible.

*Figure 5.* A sample of *Blender*.

**App:** Element

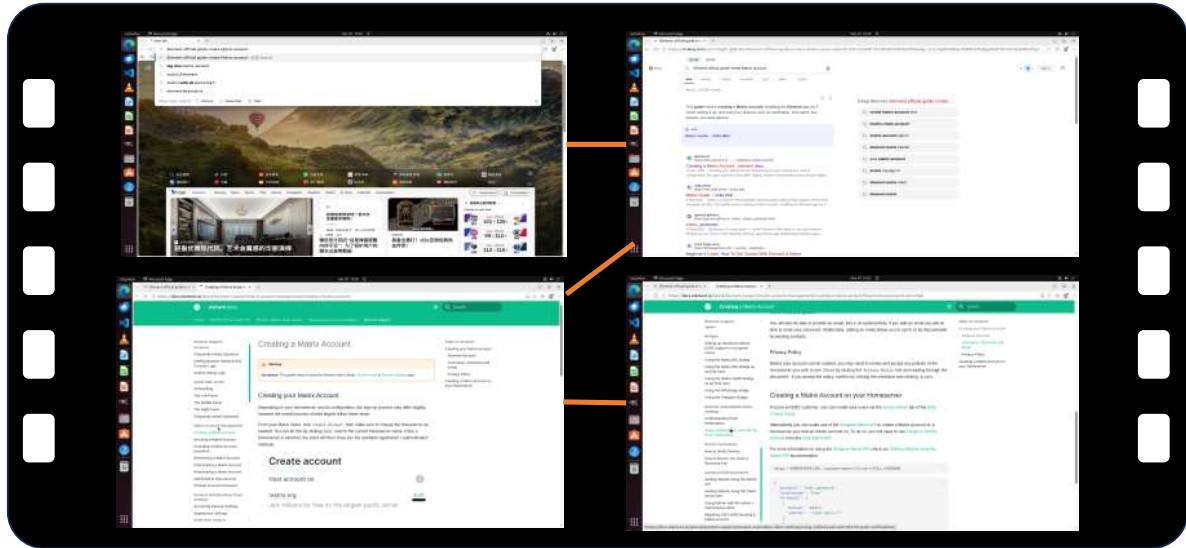

**Task Instruction**:Create a Matrix account using Element Web or Desktop, including selecting a homeserver, choosing registration methods, setting MXID and optional email, and accepting policies; also notes for admin/API-based account creation on a homeserver.

**actual_url:**https://docs.element.io/latest/element-support/matrix-account-management/creating-a-matrix-account/#username-password-and-email
**expected_url:**https://docs.element.io/latest/element-support/matrix-account-management/creating-a-matrix-account

**Steps:**
- Step_1: Open your Element Web or Element Desktop client and click Create Account.
- Step_2: Click Edit next to the current homeserver name and select the homeserver you want to join.
- Step_3: After selecting a homeserver, review the available registration/authentication methods and choose the desired method.
- Step_4: If you choose an external service, follow the provider's login flow and be aware that the external account may be tied exclusively to this Matrix account.
- Step_5: Choose a Matrix ID (MXID) in the format @username:homeserver.com; note that MXID cannot be changed later, though a display name can be changed.
- Step_6: (Optional) Provide an email address to enable password resets and optional discovery of the account by existing contacts.
- Step_7: If prompted, review the Privacy Policy for the homeserver and click Accept to proceed with account creation.
- Step_8: Complete the sign-up process and sign in to your new Matrix account.
- Step_9: If you admin a homeserver, you can create accounts via the EMS Server Admin panel or the Synapse Admin API using the provided endpoints and payloads

*Figure 6.* A sample of *Element*.

## App: Godot

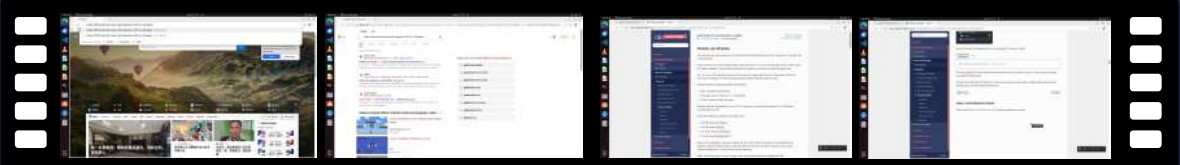

**Task Instruction**:Create and integrate a Heads-Up Display (HUD) in a Godot 2D game, with ScoreLabel, Message, StartButton, and MessageTimer, including font setup, layout anchors, signals, and game-state interactions.

**actual_url:**https://docs.godotengine.org/en/stable/getting_started/first_2d_game/06.heads_up_display.html
**expected_url:**https://docs.godotengine.org/en/stable/getting_started/first_2d_game/06.heads_up_display.html

**Steps:**
- Step_1: Create a new scene, add a CanvasLayer node named HUD to serve as the UI overlay, and add child nodes ScoreLabel, Message, StartButton, and MessageTimer.
- Step_2: Under HUD, ensure the children are ScoreLabel (Label), Message (Label), StartButton (Button), and MessageTimer (Timer).
- Step_3: Configure ScoreLabel: set initial text to 0, set Horizontal and Vertical Alignment to Center, load the Xolonium-Regular.ttf font via Theme Overrides > Fonts, and set font size to 64 via Theme Overrides > Font Sizes; use the Center Top anchor preset.
- Step_4: Configure Message: set text to 'Dodge the Creeps!', center alignments, enable Word autowrap, set Size X to 480 under Control - Layout/Transform, and use Center anchor preset.
- Step_5: Configure StartButton: set text to 'Start', set Size X to 200 and Size Y to 100, use Center Bottom anchor preset, and set Position Y to 580.
- Step_6: On MessageTimer, set Wait Time to 2 and One Shot to On.
- Step_7: Add a script to HUD implementing a start_game signal; declare the signal and ensure the project is rebuilt so the editor recognizes it.
- Step_8: Implement show_message(text) in HUD to set the Message text, show the Message, and start the MessageTimer.
- Step_9: Implement show_game_over() in HUD to display 'Game Over', wait for the MessageTimer timeout, then show 'Dodge the Creeps!' for 1 second, and finally reveal the StartButton.
- Step_10: Implement update_score(score) in HUD to update ScoreLabel with the given score.
- Step_11: Connect StartButton pressed() and MessageTimer timeout() signals to HUD; implement on_start_button_pressed() to hide StartButton and emit start_game, and on_message_timer_timeout() to hide the Message.
- Step_12: Instance the HUD scene in Main and connect HUD.start_game to Main.new_game(); verify a green connection appears next to the function in the script.
- Step_13: In new_game(), call HUD.update_score(score) and HUD.show_message('Get Ready').
- Step_14: In game_over(), call HUD.show_game_over().
- Step_15: In _on_score_timer_timeout(), call HUD.update_score(score) to keep the HUD in sync with the score.
- Step_16: Remove any automatic start call from _ready() if present, then run the project to test and fix any issues.
- Step_17: To clean up between games, assign mobs to a 'mobs' group and, in Main.new_game(), call get_tree().call_group('mobs','queue_free') to remove existing creep.

*Figure 7.* A sample of *Godot.*

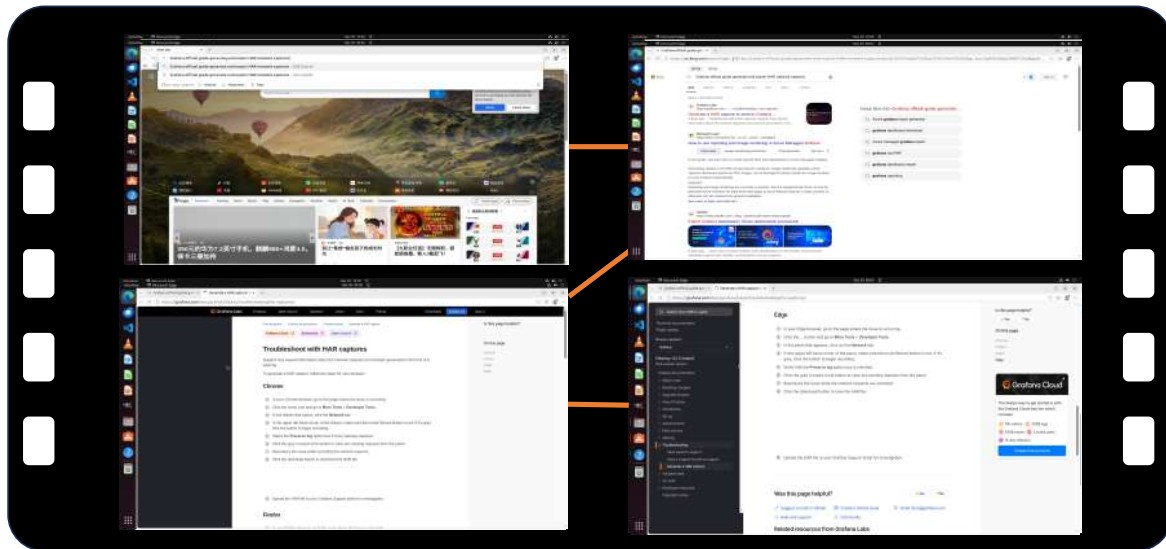

**App:** Grafana

**Task Instruction**:Generate and export HAR network captures from the browser's Developer Tools for troubleshooting Grafana issues.

**actual_url:**https://grafana.com/docs/grafana/latest/troubleshooting/har-captures/
**expected_url:**https://grafana.com/docs/grafana/latest/troubleshooting/har-captures

**Steps:**
- Step_1: In your Edge browser, go to the page where the issue is occurring.
- Step_2: Click the … button and go to More Tools > Developer Tools.
- Step_3: In the panel that appears, click on the Network tab.
- Step_4: In the upper left hand corner of the panel, make sure the round Record button is red. If it's grey, click the button to begin recording
- Step_5: Verify that the Preserve log option box is checked.
- Step_6: Click the grey crossed circle button to clear any existing requests from the panel.
- Step_7: Reproduce the issue while the network requests are recorded.
- Step_8: Click the download button to save the HAR file.
- Step_9: Upload the HAR file to your Grafana Support ticket for investigation.

*Figure 8.* A sample of *Grafana*.

App: Idea

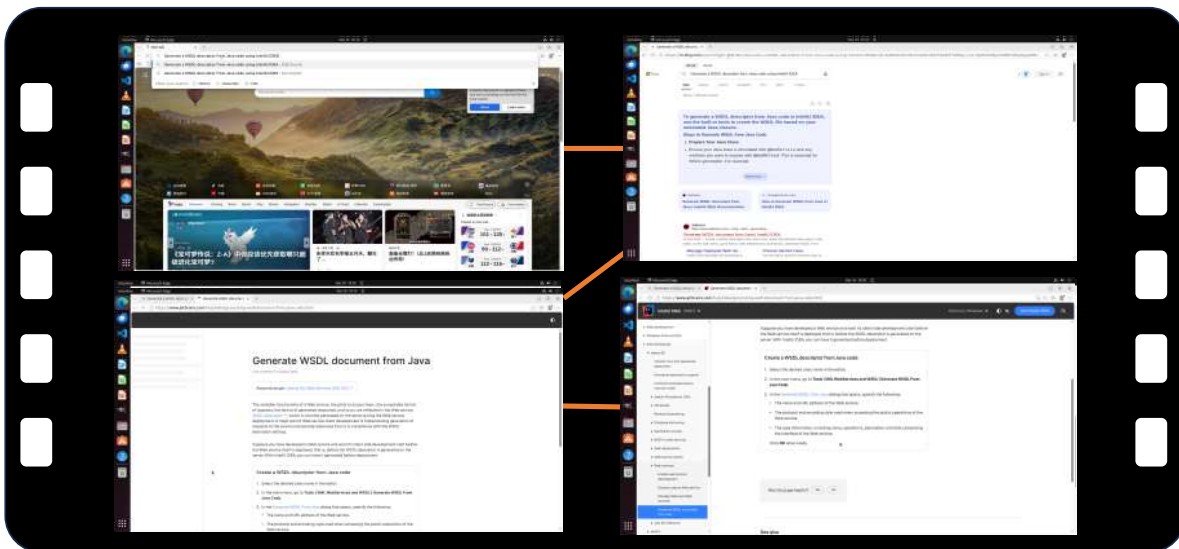

**Task Instruction**:Generate a WSDL descriptor from Java code using IntelliJ IDEA by selecting a class and using the Generate WSDL From Java Code action.

**actual_url:**https://www.jetbrains.com/help/idea/generating-wsdl-document-from-java-code.html
**expected_url:**https://docs.godotengine.org/en/stable/getting_started/first_2d_game/06.heads_up_display.html

**Steps:**
- Step_1: Select the desired Java class in the editor.
- Step_2: From the main menu, go to Tools | XML WebServices and WSDL | Generate WSDL From Java Code.
- Step_3: In the Generate WSDL From Java dialog, specify the Web service name and URL, the protocol and encoding style, and the type information (name, operations, parameters and data) of the Web service interface.
- Step_4: Click OK to generate the WSDL descriptor.

*Figure 9.* A sample of *Idea*.

**App:** Netbeans

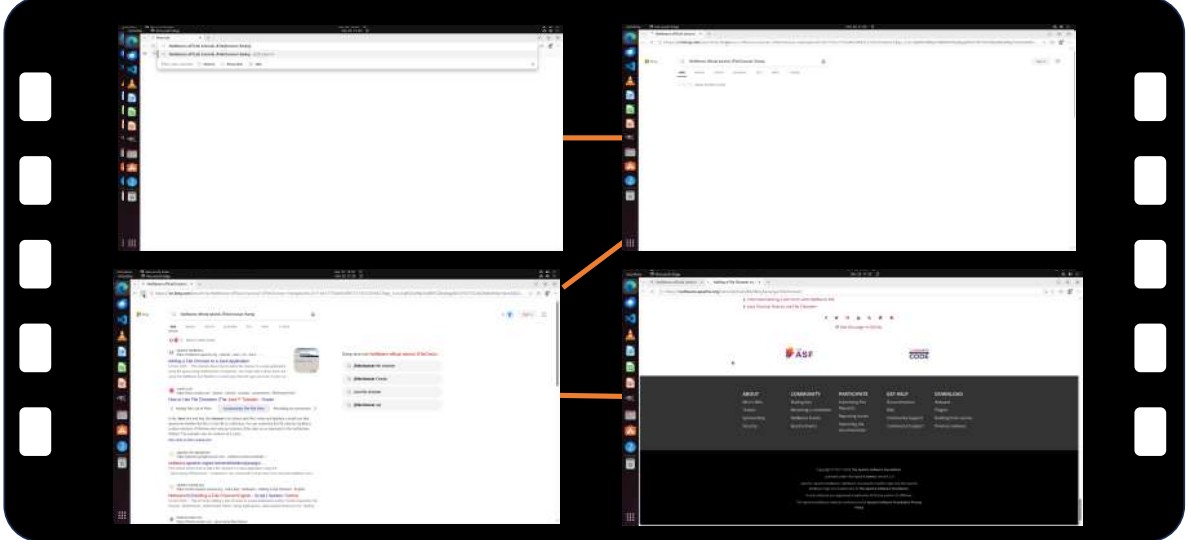

**Task Instruction**:Add and configure a JFileChooser in a Java Swing application using NetBeans GUI Builder, including implementing an open action and a file filter to load a .txt file into a TextArea.

**actual_url:**https://netbeans.apache.org/tutorial/main/kb/docs/java/gui-filechooser/
**expected_url:**https://netbeans.apache.org/tutorial/main/kb/docs/java/gui-filechooser

**Steps:**
- Step_1: Create a new Java Application project named JFileChooserDemo in NetBeans.
- Step_2: Create the application form by adding a JFrameForm named JFileChooserDemo in the appropriate package.
- Step_3: Add a Menu Bar with Open and Exit items to the frame.
- Step_4: Add a JTextArea to the form and rename its variable to textarea.
- Step_5: Add a JFileChooser component to the form and rename its variable to fileChooser.
- Step_6: Set the JFileChooser's dialogTitle property to 'This is my open dialog'.
- Step_7: Implement the OpenActionPerformed method to show the open dialog and, if a file is chosen, read its contents into the textarea.
- Step_8: Configure a file filter by opening the fileChooser's fileFilter property and selecting Custom Code.
- Step_9: Create an inner class MyCustomFilter that extends FileFilter to accept only directories or files ending with '.txt' and describe it as 'Text documents (*.txt)'.
- Step_10: Apply the custom filter in the fileChooser so that only .txt files are shown.
- Step_11: Run the application, choose Open from the File menu to test loading a .txt file into the textarea, and choose Exit to close the application.
- Step_12: Explore additional Swing components and dialogs such as ColorChooser or OptionPane as next steps

*Figure 10.* A sample of *Netbeans*.

## App: Notepad++

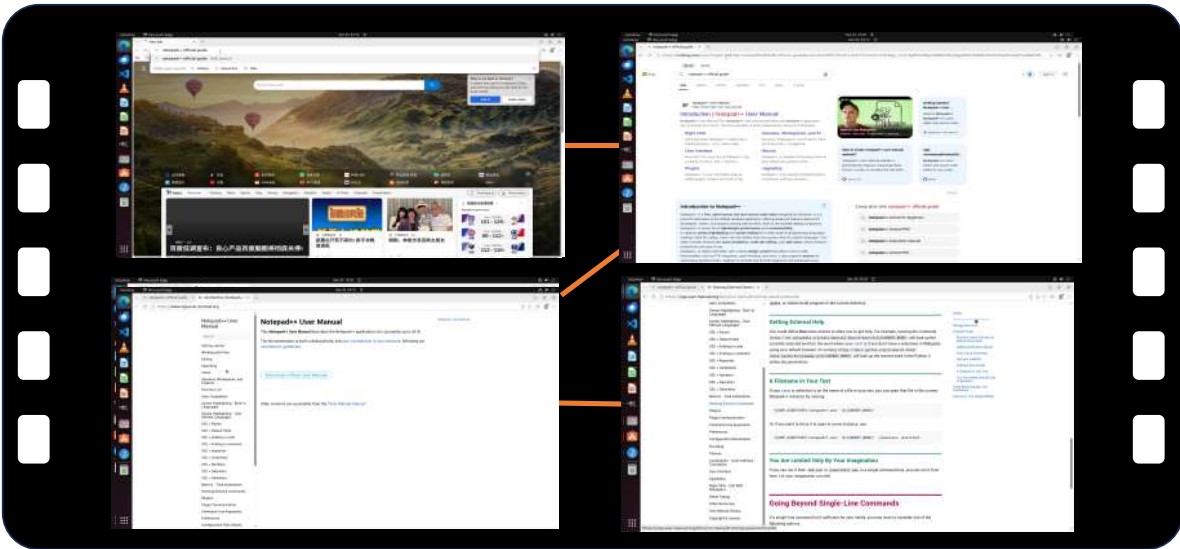

**Task Instruction**:Use Notepad++ Run… to execute commands, save and manage Run entries, and run commands with examples.

**actual_url:**https://npp-user-manual.org/docs/run-menu/#running-saved-commands
**expected_url:**https://npp-user-manual.org/docs/run-menu

**Steps:**
- Step_1: Open Notepad++ and select Run > Run… to open the Run dialog.
- Step_2: In the Program to Run field, type a command or use the ... button to browse for an executable to run.
- Step_3: If needed, prefix the command with cmd.exe /c to run it in a Windows command prompt, or cmd.exe /k to keep the window open after execution; specify powershell.exe for PowerShell commands.
- Step_4: If the command is not in PATH, provide a full path (enclose paths with spaces in quotes, e.g. \"C:\\Program Files\\App\\app.exe\" \"arguments\").
- Step_5: Use the pulldown to reuse previous commands and click Run to execute; click Save… to save the command as a named Run entry and optionally assign a keyboard shortcut.
- Step_6: Click Cancel or the X to exit the dialog without running.
- Step_7: To run saved commands, select them under Run… in the menu; use Run > Modify Shortcut / Delete Command to manage shortcuts or remove commands.
- Step_8: See examples: run the current file with its default association using \"$(FULL_CURRENT_PATH)\"; open a file in a browser or other app by saving a corresponding command (e.g., a browser executable with \"$(FULL_CURRENT_PATH)\").
- Step_9: Be mindful of security: running external programs can affect your PC; only run commands from trusted sources

*Figure 11.* A sample of *Notepad++*.

## App: Odoo

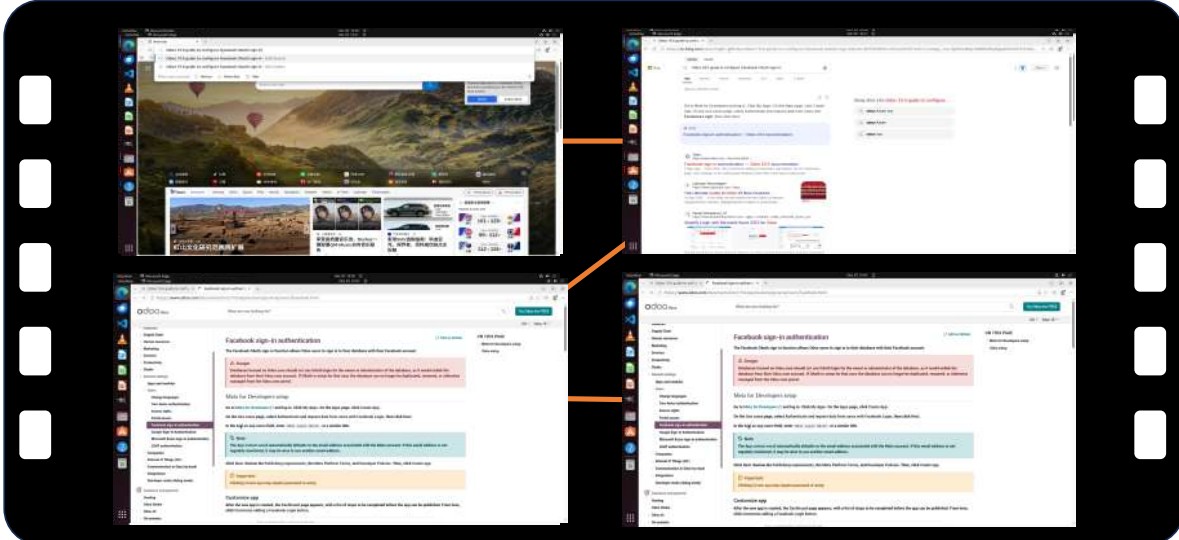

**Task Instruction**:Configure Facebook OAuth sign-in for Odoo by creating a Facebook app, adjusting settings, and wiring the App ID into Odoo OAuth Providers.

**actual_url:**https://www.odoo.com/documentation/19.0/applications/general/users/facebook.html
**expected_url:**https://www.odoo.com/documentation/19.0/applications/general/users/facebook.html

**Steps:**
- Step_1: Go to Meta for Developers and log in.
- Step_2: Click My Apps, then Create App. On the Use cases page, select Authenticate and request data from users with Facebook Login, then click Next.
- Step_3: In Add an app name, enter 'Odoo Login OAuth' (or a similar title). Note that the App contact email defaults to the Meta account email and may require a password re-entry.
- Step_4: Review the Publishing requirements and Developer Policies, then click Create app.
- Step_5: In the Dashboard, click Customize and add a Facebook Login button.
- Step_6: On Customize > Settings, set Valid OAuth Redirect URIs to https://<odoo base url>/auth_oauth/signin (replace <odoo base url> with your database URL) and click Save changes.",
- Step_7: In App settings > Basic, set the Privacy Policy URL to https://www.odoo.com/privacy, upload an app icon, set User data deletion URL to https://www.odoo.com/documentation/17.0/administration/odoo_accounts.html, and set Category to Business and pages; Save changes.",
- Step_8: After creating the app, copy the App ID from the dashboard for use in the next steps.",
- Step_9: In the Meta Developers dashboard, click Publish. Depending on the connected Facebook account, you may need to complete additional verification steps and then publish.",
- Step_10: In Odoo, enable Developer mode (Settings) and go to Settings app → Integrations → OAuth Providers. Click Facebook Graph and enter the App ID in the Client ID field, then tick Allowed.",
- Step_11: Sign in to the Odoo database to verify the Facebook OAuth sign-in works as expected."

*Figure 12.* A sample of *Odoo*.

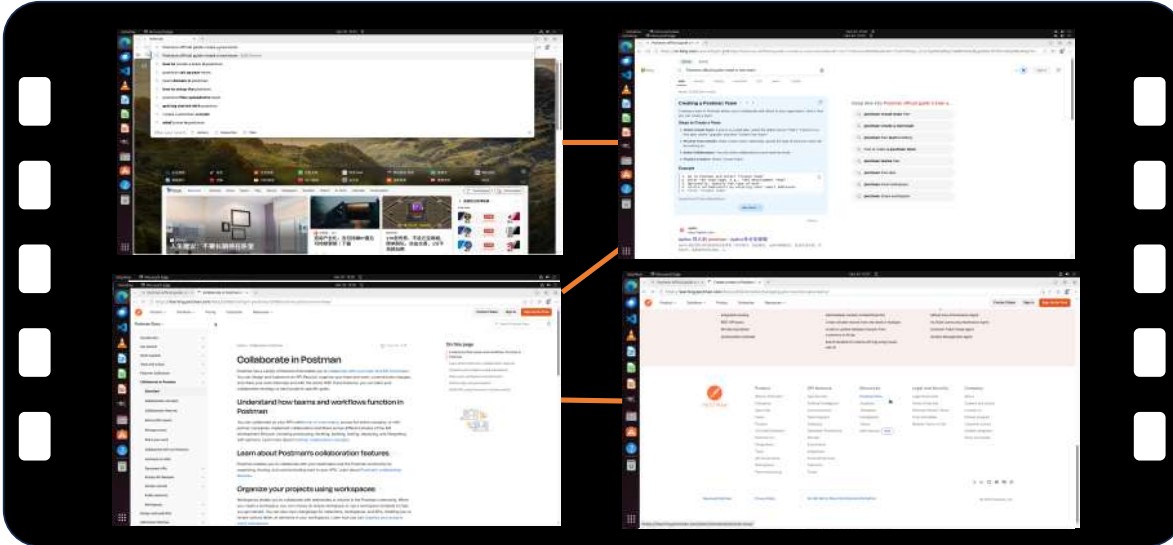

App: Postman

**Task Instruction**:Create a new team in Postman, including naming the team, optionally specifying the work type, and inviting collaborators, with subsequent team settings adjustments.

**actual_url:**https://learning.postman.com/docs/administration/managing-your-team/create-teams/
**expected_url:**https://learning.postman.com/docs/administration/managing-your-team/create-teams

**Steps:**
- Step_1: Click the Team (or Upgrade) option in the Postman interface, then select Create team (or Create Free team on the free plan).
- Step_2: Enter a team name and, optionally, specify the type of work the team will handle.
- Step_3: Click Continue.
- Step_4: (Optional) Invite collaborators by email or by adding people from a contacts file.
- Step_5: Click Create Team.

*Figure 13.* A sample of *Postman*.

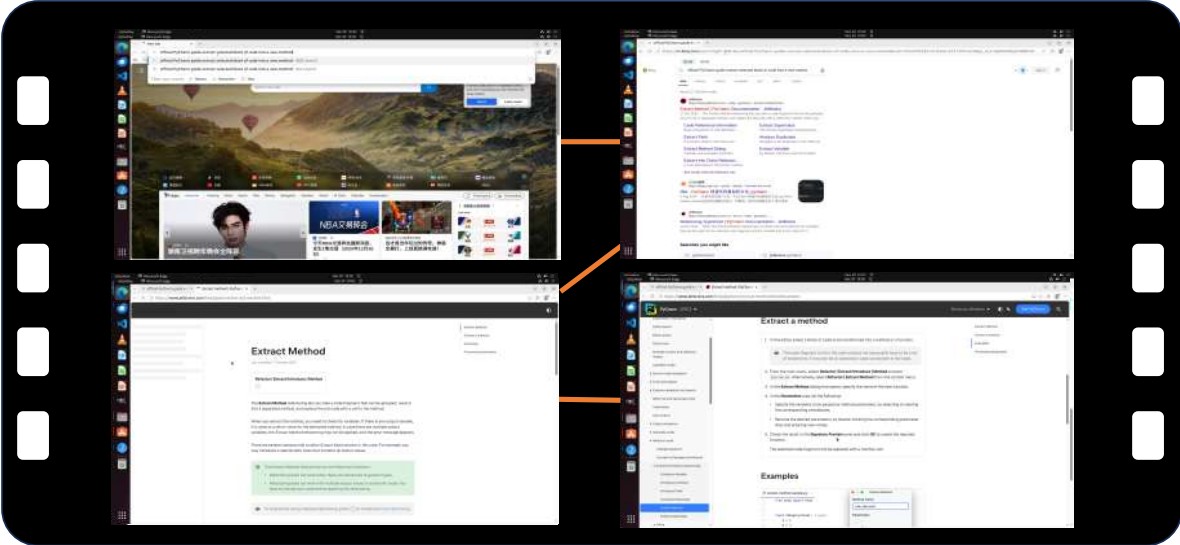

| App: Pycharm |
| --- |

**Task Instruction:**Extract a selected block of code into a new method in PyCharm, including naming the method and configuring parameters, so the code fragment is replaced by a method call.

**actual_url:**https://www.jetbrains.com/help/pycharm/extract-method.html#examples
**expected_url:**https://www.jetbrains.com/help/pycharm/extract-method.html

**Steps:**
- Step_1: In the editor, select a block of code to be transformed into a method or a function.
- Step_2: From the main menu, select Refactor | Extract/Introduce | Method or press Ctrl+Alt+0M. Alternatively, select Refactor | Extract Method from the context menu.
- Step_3: In the Extract Method dialog that opens, specify the name of the new function.
- Step_4: In the Parameters area, specify the variables to be passed as method parameters by selecting or clearing the corresponding checkboxes; rename parameters by double-clicking the parameter lines and entering new names.
- Step_5: Check the result in the Signature Preview pane and click OK to create the required function; the selected code fragment will be replaced with a function call.

*Figure 14.* A sample of *PyCharm.*

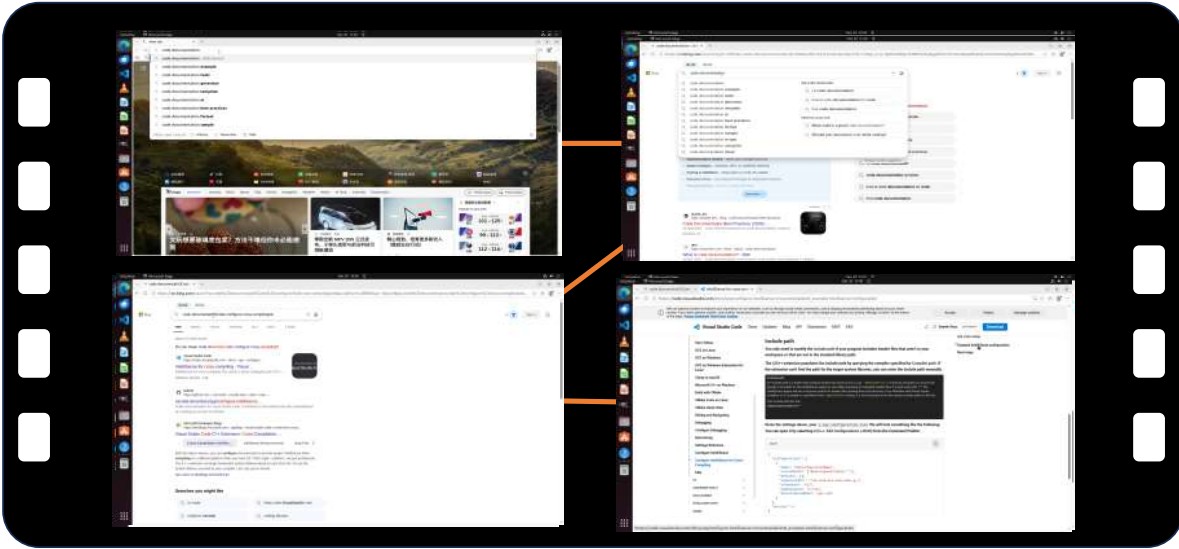

App: VS Code

**Task Instruction:**Configure VS Code to emulate the target architecture for cross-compiling by setting compilerPath and IntelliSenseMode (and optionally includePath) in c\_cpp\_properties.json.

**actual_url:**https://code.visualstudio.com/docs/cpp/configure-intellisense-crosscompilation#_example-intellisense-configuration
**expected_url:**https://code.visualstudio.com/docs/cpp/configure-intellisense-crosscompilation

**Steps:**
- Step_1: In VS Code, open the Command Palette (Ctrl+Shift+P) and select \"C/C++: Edit Configurations (UI)\".
- Step_2: In the UI, set the Compiler path to your cross-compiler (e.g., /usr/bin/arm-none-eabi-g++) and set IntelliSense mode to the target architecture (e.g., gcc-arm).
- Step_3: If necessary, modify the Include path to point to headers not found automatically (manually add paths).
- Step_4: Open the JSON version by selecting \"C/C++: Edit Configurations (JSON)\" from the Command Palette.
- Step_5: Ensure the configuration includes fields like compilerPath, IntelliSenseMode, includePath, cStandard, cppStandard (as shown in the example).
- Step_6: Save settings and verify that IntelliSense reflects the target architecture for cross-compilation.

*Figure 15.* A sample of *VSCode.*

**App:** Zotero

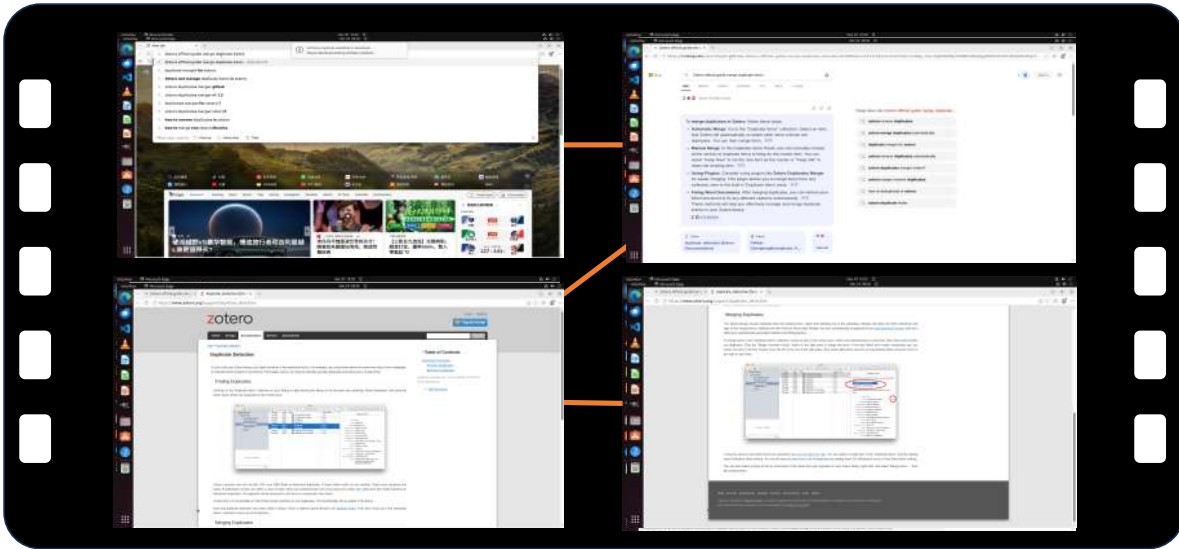

**Task Instruction**:Locate and merge duplicate items in Zotero using the Duplicate Items view, including selecting duplicates, choosing a master item, and merging, with optional guidance for merging via context menu.

**actual_url:**https://www.zotero.org/support/duplicate_detection
**expected_url:**https://www.zotero.org/support/duplicate_detection

**Steps:**
- Step_1: Open Zotero and click the 'Duplicate Items' collection in the left pane, or right-click your library and select 'Show Duplicates' to display potential duplicates in the center pane.
- Step_2: Review the items listed as duplicates; Zotero uses title, DOI, and ISBN to identify duplicates, and may also consider year and overlapping authors when necessary.
- Step_3: In the Duplicate Items view, select one of the duplicate items in the center pane; Zotero will automatically co-select the other items it thinks are duplicates.
- Step_4: Click the 'Merge <number> Items' button in the right pane to merge the selected items into a single item.
- Step_5: If the item fields don't match completely, choose one item to be the 'master' from the list at the top of the right pane, then use the icons to the right of each field to select alternative versions for mismatched fields.
- Step_6: (Optional) Sort the items by Title to make it easier to see duplicates; you can select a single item in the 'Duplicate Items' view by holding down Alt/Option while clicking.
- Step_7: To deselect an item from a set of duplicates, hold down Ctrl (Windows/Linux) or Cmd (Mac) while clicking.
- Step_8: You can also select two or more items of the same item type anywhere in your Zotero library, right-click, and choose 'Merge Items…' from the context menu

*Figure 16.* A sample of *Zotero*.

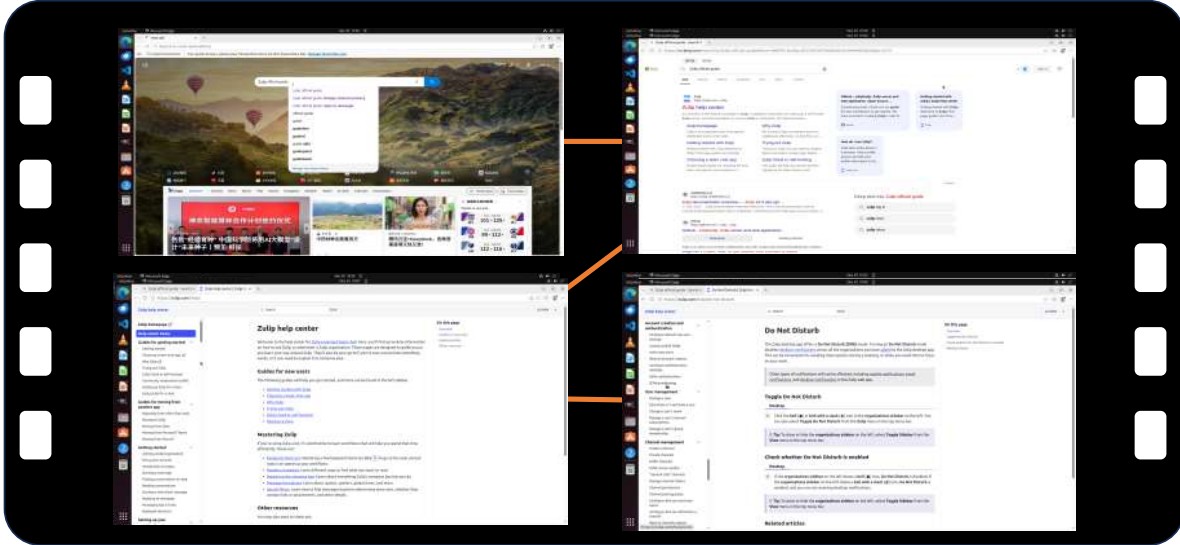

**App:** Zulip

**Task Instruction:** Enable or disable Do Not Disturb (DND) in the Zulip desktop app and verify its status.

**actual_url:** https://zulip.com/help/do-not-disturb
**expected_url:** https://zulip.com/help/do-not-disturb

**Steps:**
- Step_1: In the Zulip desktop app, click the bell icon on the left organizations sidebar to toggle Do Not Disturb, or use the Toggle Do Not Disturb option in the Zulip menu in the top bar.
- Step_2: To verify the DND status, check the icon in the organizations sidebar: a regular bell indicates DND is disabled, while a bell with a slash indicates DND is enabled and desktop notifications are suppressed.
- Step_3: (Optional) To show or hide the organizations sidebar, select View > Toggle Sidebar from the top menu.

*Figure 17.* A sample of *Zulip.*

**Error Type：Imprecise Localization**

**Task Instruction:** Enable and use preview snapping in Blender's Video Sequencer to snap images to borders, center, or other strips, including enabling/disabling snapping and temporary Ctrl-based snapping.

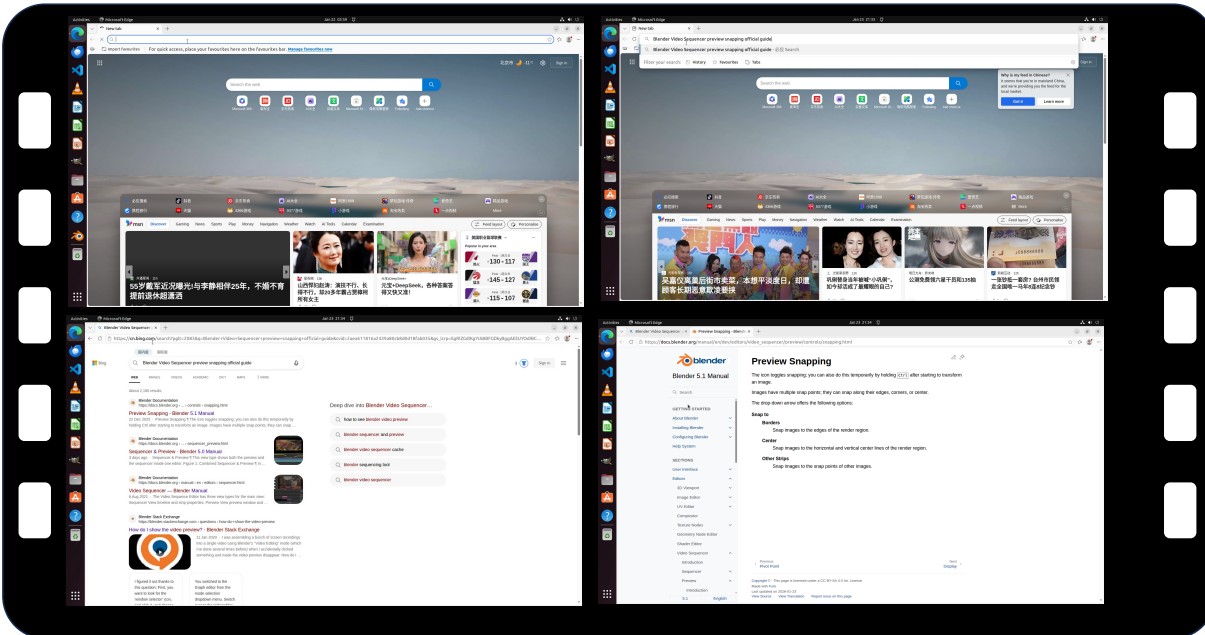

**actual_url:** https://docs.blender.org/manual/en/dev/editors/video_sequencer/preview/controls/snapping.html
**expected_url:** https://docs.blender.org/manual/en/latest/compositing/types/filter/sun_beams.html

*Figure 18.* An error case of imprecise localization.

### Error Type：Non-official Reference

**Task Instruction:** Explain how to use the Hook modifier in Blender to deform a mesh by linking it to another object, including vertex assignment, falloff, and an example with an Empty.

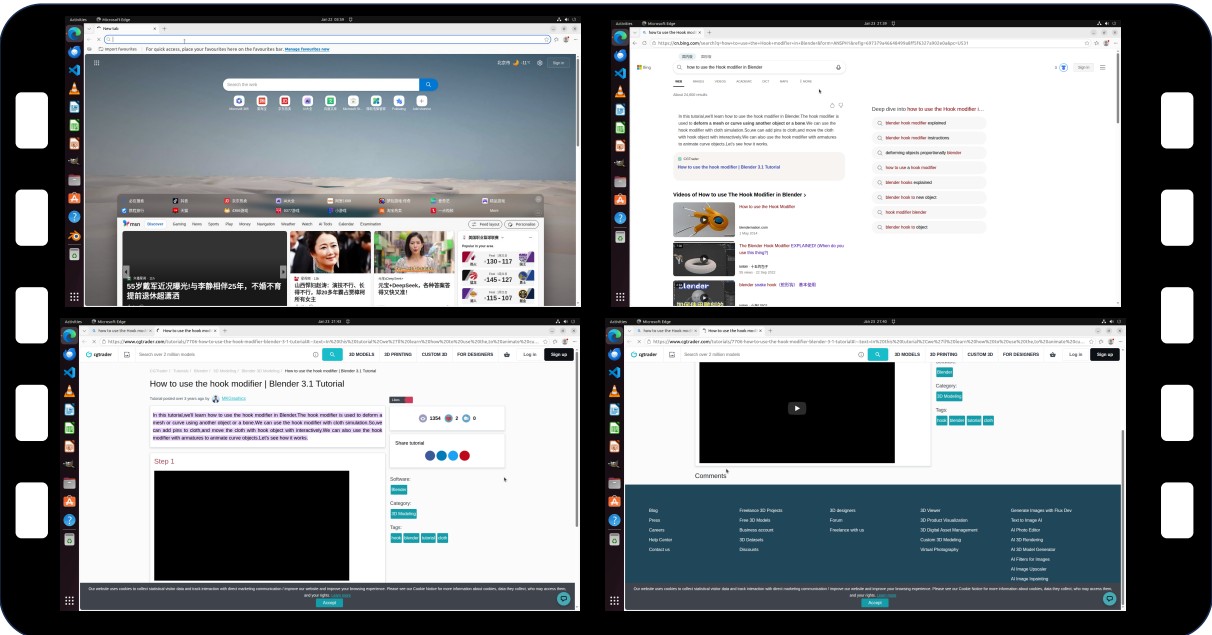

**actual_url:** https://www.cgtrader.com/totorials/7706-how-to-use-the-hook-modifier-blender-3-1-tutorial#:~:text=...
**expected_url:** https://docs.blender.org/manual/en/latest/modeling/modifiers/deform/hooks.html

*Figure 19.* An error case of Non-official Reference.

Error Type： Execute Before Retrieval

**Task Instruction:** Guide to use Blender's Knife Project tool to project the outline of non-editing objects onto meshes in Edit Mode and cut them accordingly.

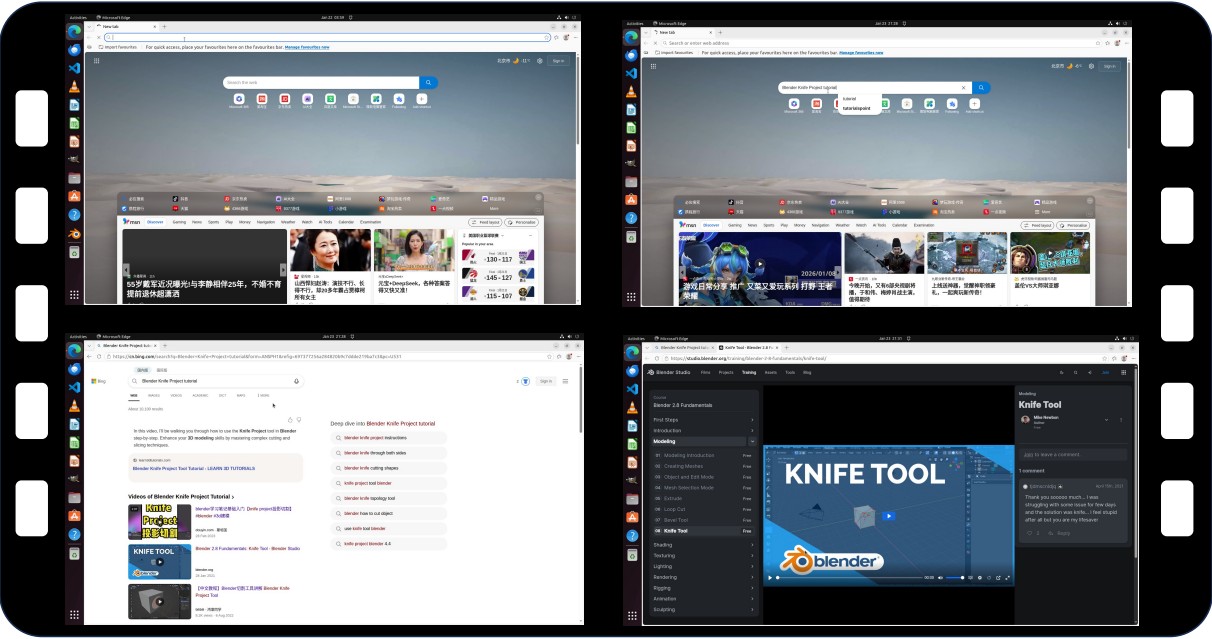

**actual_url:** https://studio.blender.org/training/blender-2-8-fundamentals/knife-tool
**expected_url:** https://docs.blender.org/manual/en/latest/movie_clip/tracking/clip/marker.html

*Figure 20.* An error case of execute before retrieval.

### Error Type: Action Grounding Failure

**Task Instruction:** Configuring PyCharm for testing by creating a Test Sources Root and marking a directory to store test code.

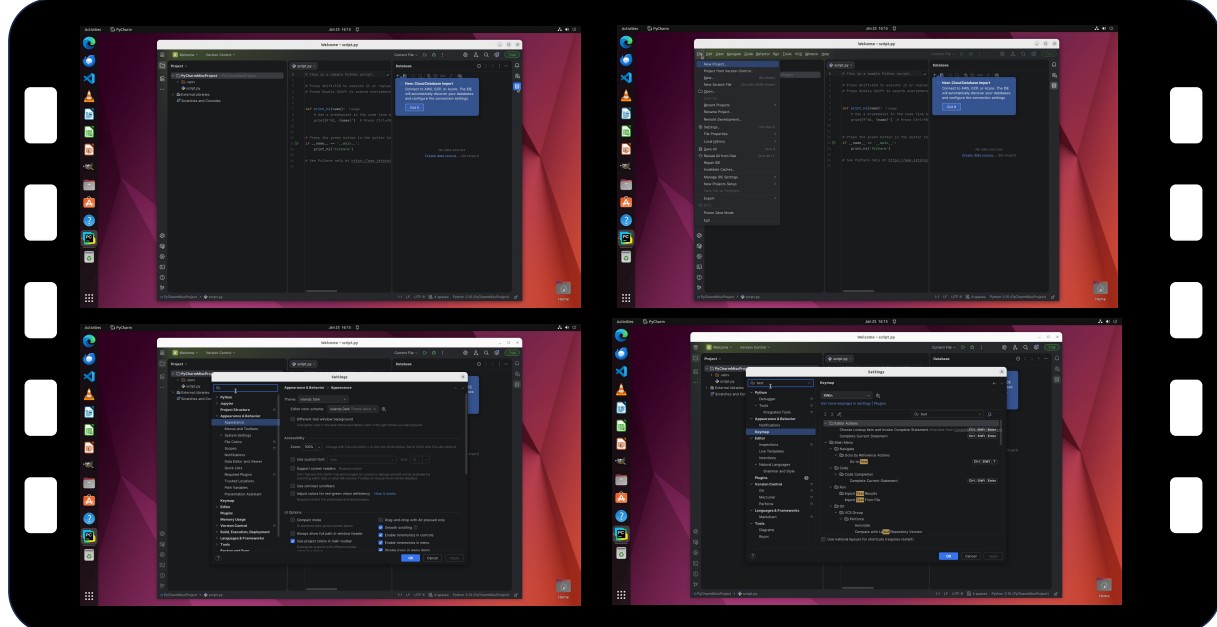

**Steps**
- **Step_1**: In the Project tool window (Alt+01), create a new directory in which you will store your test code.
- **Step_2**: Right-click the new directory and select Mark Directory As | Test Sources Root.
- **Step_3**: Ensure the folder is marked with the Test Root icon, indicating it is configured as a Test Sources Root.

**Explanation**
- **Step 1 is completed**: the agent used the Project tool window context menu to create a new directory named 'test'.
- **Step 2 is completed**: the agent right-clicked the 'test' directory and selected Mark Directory As Test Sources Root.
- **Step 3 is not clearly completed** because the trajectory provides no verification that the folder shows the Test Root icon/visual marking after the action.

*Figure 21.* An error case of action grounding failure.

Error Type： Context Misidentification

**Task Instruction:** Configuring PyCharm for testing by creating a Test Sources Root and marking a directory to store test code.

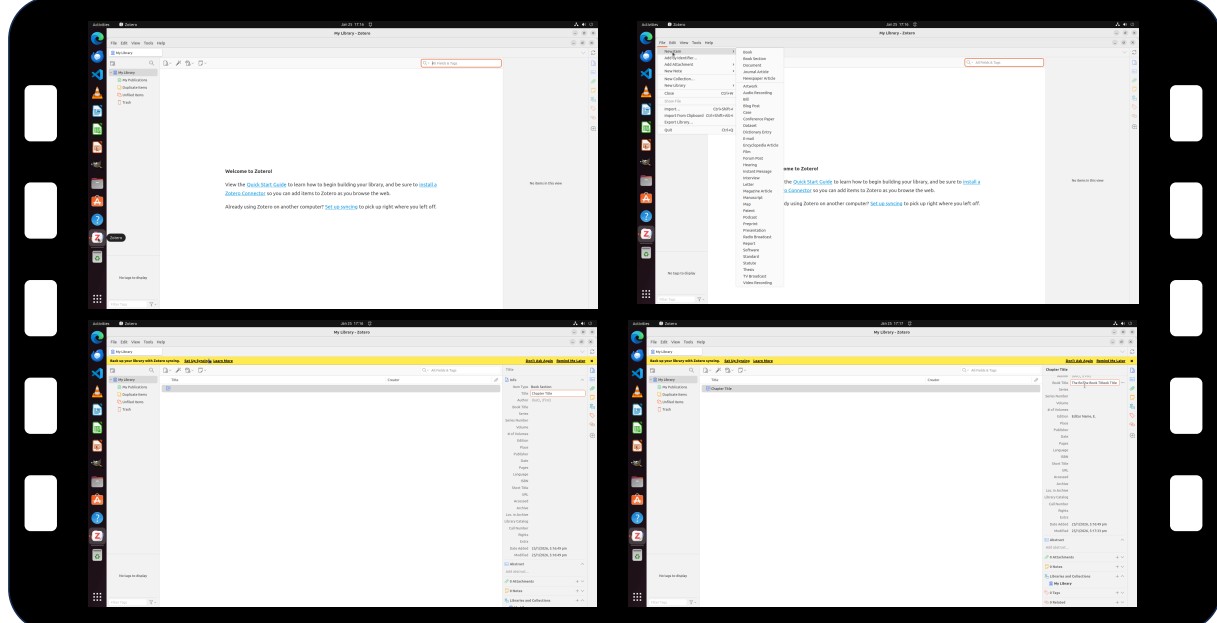

**Steps**
- **Step_1:** Click the green new item button and select 'Book Section'.
- **Step_2:** Enter the chapter title in the Title field and the book title in the Book Title field.
- **Step_3:** To add an editor, click the '+' sign on the author line in the right column to create an additional author line.
- **Step_4:** Click the 'Author' label on the newly created line and change it to editor.

**Explanation**
- **Step 1 completed**: agent clicked the green New Item button and selected 'Book Section' (Steps 2\u20133).
- **Step 2 completed**: agent entered a chapter title in Title ('The Quick Python Book') and a book title in Book Title ('Python Tricks').
- **Steps 3 & 4 not completed as specified**: the agent changed an existing Author role to Editor but did not click the '+' to add an additional creator line and then change that new line's role to Editor.

*Figure 22.* An error case of context misidentification.

## D. Semantic Retrieval Evaluation Metric

To further strengthen the evaluation of the retrieval phase, we complement the original URL-based metrics with an additional semantic relevance metric. Specifically, we compute semantic similarity between the retrieved documents and the corresponding ground-truth references using embedding-based representations. As shown in Table 8, the semantic relevance metric exhibits a similarity score of 0.653 with the URL-based evaluation results, which indicates that the URL-based metrics are reasonably aligned with semantic relevance, supporting the validity of our original evaluation protocol.

*Table 8.* Performance comparison of various GUI agents with Sentence-BERT.

| **Models** | Sentence-BERT |
| --- | --- |
| GELab-Zero-4B-preview | 18.62 |
| UI-TARS-1.5-7B | 26.23 |
| MAI-UI-8B | 32.19 |
| Aguvis-7B | 27.27 |
| GUI-Owl-7B | 10.04 |
| Qwen3-VL-8B | 37.88 |

## E. Additional Baselines

To address potential limitations in the diversity of baselines, we extend our experimental setup by incorporating both a larger-scale model and an additional agent framework. Specifically, we include the Qwen3-VL-32B as a higher-parameter baseline and integrate the AutoGen framework based on Qwen3-VL-7B as an alternative agent-based system. The results of these additional experiments are shown in Table 9. Overall, we observe that while larger-scale models demonstrate improved capabilities in certain aspects, the relative trends and conclusions reported in the main paper remain consistent. This suggests that our findings are not limited to a specific model scale or agent framework.

*Table 9.* Performance of larger-scale model and agentic framework.

| **Models** | TUI | HPP | TCR |
| --- | --- | --- | --- |
| Qwen3-VL-32B | 45.66 | 55.74 | 6.21 |
| AutoGen (Qwen3-VL-8B) | 41.21 | 53.63 | 5.13 |

## F. Prompt Template

*Table 10.* Prompt Template for LLM as a Judge.

---

**Prompt Template for LLM as a Judge.**

You are an expert evaluator of GUI task execution. Based on the Ground Truth steps, evaluate how many of them the GUI Agent has completed.

**Task description:**
{task_description}

**Ground Truth steps:**
{ground_truth_steps}

**GUI Agent trajectory:**
{trajectory_text}

**Guidelines:**

- Count only steps the agent clearly completed or was very close to completing successfully.

- If the agent attempted but did not succeed, do NOT count it.

- If the agent performed extra detail beyond Ground Truth, count only the corresponding Ground Truth step(s).

**Output format:**

```
{
  "steps": <number of completed Ground Truth steps>,
  "Explanation": "<concise but specific explanation>"
}
```

---

*Figure 23.* Prompt for GUI Content Classification.

---

**Prompt for Document Content Classification**

You are an assistant that classifies content based on specific criteria. Your task is to evaluate whether a given piece of content serves as a tutorial specifically related to graphical user interfaces (GUI), such as web applications, desktop applications, or operating systems.
The content qualifies as a GUI-related tutorial if it meets the following conditions:

1. It includes a task description outlining what needs to be achieved.

2. It provides clear step-by-step instructions for interacting with a GUI, such as:
   - Step 1: Open the application.
   - Step 2: Navigate to the settings menu.

Given the URL and context, determine whether the content is a GUI-related tutorial. Output 1 if it is a GUI-related tutorial and 0 otherwise. Provide only the number as the output.

**User Input:**

- URL: {url}

- Context: {context}

---

