# OpenReview forum: "DocOS: Towards Proactive Document-Guided Actions in GUI Agents"
_ICML.cc/2026/Conference — ICML 2026 regular_

### Official Review · Reviewer_bp1B · 2026-03-02

**Soundness:** 3
**Presentation:** 3
**Significance:** 2
**Originality:** 2
**Overall Recommendation:** 3
**Confidence:** 3

**Summary:**

This paper introduces DocOS, a benchmark for proactive document-guided actions in GUI agents. The task is decomposed into two phases: proactive knowledge retrieval, where the agent autonomously locates relevant online documentation, and document-grounded execution, where the agent completes GUI tasks conditioned on the retrieved document.

DocOS contains 817 desktop tasks across 20 applications with varying difficulty levels. Evaluation metrics include Target URL Inclusion (TUI) and Hierarchical Path Progress (HPP) for retrieval, and Task Completion Rate (TCR) for execution. Experiments on multiple multimodal foundation models show low task completion rates and performance degradation with increasing task length.

**Compliance With Llm Reviewing Policy:**

Affirmed.

**Final Justification:**

Taking into account the opinions of the other reviewers and rebuttals of authors, I decide to maintain my score.

**Key Questions For Authors:**

1. Could the authors clarify the distribution of tasks across applications (e.g., the proportion from PyCharm) and discuss how this scale and concentration affect the generality and representativeness of DocOS as a benchmark?

**Limitations:**

Yes. The authors provide an adequate discussion of limitations. The work focuses on benchmark construction and empirical evaluation, and no significant negative societal impact is apparent.

**Strengths And Weaknesses:**

1. **Strengths**

(1) Clear task formulation. The paper introduces the concept of proactive document-guided actions and explicitly decomposes the problem into proactive knowledge retrieval and document-grounded execution, providing a clear and structured evaluation setting.

(2) Well-designed benchmark protocol. DocOS contains 817 desktop tasks across 20 applications with difficulty stratification and distinct metrics (TUI, HPP, TCR) for different phases, along with oracle document ablations.

(3) Comprehensive empirical evaluation. The study evaluates multiple multimodal foundation models and analyzes performance degradation with increasing task length, offering a systematic empirical characterization.

(4) Timely and practically relevant setting. The benchmark targets long-tail, feature-rich GUI tasks that require consulting official documentation, addressing an emerging capability gap in GUI agents.



2. **Weaknesses**

(1) Scope of methodological contribution.
The paper mainly introduces a benchmark and evaluation protocol, without proposing new learning algorithms or training strategies. This may limit its methodological advancement relative to typical ICML main-track contributions.

(2) Retrieval evaluation metrics.
The retrieval phase relies primarily on URL-based measures (TUI and HPP). Additional evaluation of semantic relevance or document grounding would further strengthen the validation.

(3) Depth of analysis.
The empirical results are informative but largely descriptive. More in-depth analysis of long-horizon error accumulation or systematic ablations could enhance insight.

(4) Limited scale and concentration of the benchmark.
DocOS contains 817 desktop tasks across 20 applications, which is relatively modest in scale for an ICML main-track contribution centered on a new benchmark. Moreover, a substantial portion of the tasks are drawn from a single application (e.g., PyCharm), leading to potential distribution imbalance. The overall scale and diversity may be insufficient to establish a broadly representative evaluation standard at the level typically expected for ICML.

If the methodological positioning and benchmark scale are substantially strengthened, including clearer articulation of the intended methodological advances, improved analysis depth, and stronger evidence of scale and distribution diversity, I would be willing to reconsider and potentially increase my score.

---

> ### Author Rebuttal · Authors · 2026-03-31
>
> # Q1: Scope of methodological contribution.
>
> **A1**: We would like to clarify that the primary contribution of this work lies in the **formalization of a new interaction paradigm**, **Proactive Document-Guided Action**, rather than solely introducing a benchmark. This paradigm is grounded in **interactive, open-web environments**, where agents must actively acquire and utilize external procedural knowledge to solve long-tailed tasks.
>
> Importantly, our paradigm differs fundamentally from existing approaches:
>
> (1) Unlike **retrieval-augmented methods**, which assume **a static and pre-collected document pool**, our setting requires **autonomous, real-time information acquisition from the open web**;
>
> (2) Unlike **general web search tasks**, which focus on **information retrieval**, our framework requires agents to **ground structured documents in dynamic web environment and turn it into executable GUI actions**.
>
> This paper provides a methodological foundation for developing **agents capable of self-updating and autonomous decision-making in dynamic, real-world environments**, opening a new research direction beyond conventional static retrieval or search-based frameworks.
>
> # Q2: Retrieval evaluation metrics.
>
> **A2**: We thank the reviewers for the suggestion. We added a semantic relevance metric for retrieval, showing a similarity of 0.653 to the URL-based metric, supporting the reasonableness of our evaluation.
>
> | Model                 | Sentence-BERT |
> | --------------------- | ------------- |
> | GELab-Zero-4B-preview | 18.62         |
> | UI-TARS-1.5-7B        | 26.23         |
> | MAI-UI-8B             | 32.19         |
> | Aguvis-7B             | 27.27         |
> | GUI-Owl-7B            | 10.04         |
> | Qwen3-VL-8B           | 37.88         |
> # Q3：Depth of analysis.
> **A3**：We appreciate the reviewer’s suggestion to move beyond descriptive results and provide a **mechanistic diagnosis** of GUI agents through error-propagation analysis.
>
> The agent’s **context management capability** plays a key role in its performance. It can successfully complete individual steps but fails to maintain state transitions over long time spans. To investigate this further, we incorporated three rounds of historical observations and actions and conducted a statistical analysis of task steps and success rates. In the setting with documents (\textbf{w/}), single-step tasks achieve nearly 100% success, whereas tasks exceeding four steps drop to nearly 0%. This indicates that **a limited observation history leads the agent to repeatedly make errors**, highlighting the need to design **agents with long-term state tracking and error-correction mechanisms** as an important direction for future improvement.
>
>  # Q4：Limited scale and concentration of the benchmark.
>
> **A4**: We apologize for any lack of clarity in our initial submission and would like to address your concerns as follows:
>
> 1. Scale
>
> We would like to clarify that 817 tasks in **DocOS** actually represent **a substantial and highly competitive scale** compared to existing high-level GUI benchmarks, such as **OSWorld (369 tasks)**[1],  **CRAB (100 tasks)**[2], and **WebArena (812 tasks)**[3].
>
> [1]Xie, T., Zhang, D., Chen, J., et al. Osworld: Benchmarking multimodal agents for open-ended tasks in real computer environments.
>
> [2] Xu, T., Chen, L., Wu, D.-J., et al. Crab: Cross-environment agent benchmark for multimodal language model agents.
>
> [3]Zhou, S., Xu, F. F., Zhu, H., et al. Webarena: A realistic web environment for building autonomous agents.
>
> 2. Distribution
>
> To clarify data distribution, Table 7 shows task counts per application; we also provide distribution across software categories.
>
> | Category  | Applications                                      | Total Tasks | Percentage |
> | --------- | ------------------- | ----------- | ---------- |
> | IDE       | PyCharm, Idea, VSCode, ... | 449         | 55.4%      |
> | Game Dev  | Blender, Godot                                    | 166         | 20.5%      |
> | API Tool  | Postman                                           | 94          | 11.6%      |
> | Research  | Zotero, Anki                                      | 47          | 5.8%       |
> | Messaging | Zulip, Element, Signal                            | 36          | 4.4%       |
> | Else      | Grafana, Metabase, Odoo, ...   | 25          | 3.1%       |
>
> DocOS prioritizes **high-complexity professional tools** like IDEs, which make up 55% of all tasks, to rigorously test GUI automation limits. For instance, PyCharm dominates the dataset as a **feature-rich ecosystem** with dense functionality—file I/O, debugging, version control, and complex configurations—allowing in-depth evaluation despite being a single app. This focus addresses the gap in existing **wide but shallow** benchmarks, making DocOS well-suited for research on **complex professional software operation guidance**.
>
> [1]Lu Y, Yang J, Shen Y, et al. Omniparser for pure vision based gui agent

---

> > ### Author Rebuttal · Reviewer_bp1B · 2026-04-01
> >
> > Thank you for the detailed rebuttal and clarifications. My questions have been largely addressed. However, concerns regarding the overall contribution and empirical limitations remain, and I am inclined to maintain my original score.

---

### Official Review · Reviewer_w5N4 · 2026-03-09

**Soundness:** 2
**Presentation:** 2
**Significance:** 2
**Originality:** 2
**Overall Recommendation:** 3
**Confidence:** 5

**Summary:**

This paper addresses the limitations of existing GUI agents, which rely solely on static knowledge acquired during training and are unable to handle long-tail tasks. It proposes a new paradigm of "proactive document-guided operation," enabling agents to proactively search online documents to solve unknown tasks, much like humans do. To this end, the authors constructed a benchmark called DocOS, containing 817 high-quality tasks covering 20 applications. The agent is required to autonomously complete two phases in a fully interactive environment: first, searching for and locating relevant official documents in a web browser; and second, understanding and executing specific GUI operations based on the retrieved documents. Experimental results reveal a dual bottleneck faced by current agents: the difficulty in accurately locating relevant information and the challenge in accurately translating retrieved instructions into actual operations.

**Compliance With Llm Reviewing Policy:**

Affirmed.

**Final Justification:**

Thank you for the later replies. After carefully analyzing the task type, I think the author's point of view is reasonable, so I updated my score.

**Key Questions For Authors:**

1. Document retrieval is a very complex research area. This paper only used the most basic methods. When the data and tasks are significantly expanded, this approach to solving the problem may not necessarily yield positive feedback. Have other methods been tried?
2. More baseline requirements should include at least more agent frameworks and agent models with a larger parameter scale.
3. To the best of my knowledge, these 7b models were clearly not trained on document understanding, and judging from their current performance during the evaluation phase, they are likely to be very poor.
4. Dividing tasks into steps is too naive; different apps have significantly different levels of page complexity.
5. I haven't seen any significant innovation in document organization and data collection, which I believe is the main reason limiting the current scale of the benchmark.
6. How can you determine if the current document is truly helpful for the current task? Many tasks that seem unrelated can have their completion rate improved through the connections between pages or through indirect experience.

**Limitations:**

No, they did not discuss the limitations of their work. Pls refer to the weakness part.

**Strengths And Weaknesses:**

Strengths：
1. Dynamic Knowledge Acquisition: The agent is no longer limited to static knowledge acquired during training and can process long-tail tasks not seen in the training data by retrieving external documents in real time, exhibiting stronger generalization ability.

2. Pioneering Evaluation System: The first benchmark test to simultaneously evaluate "document search capability" and "document-driven execution capability," filling a gap in this field.

3. Fully Interactive Environment: Evaluated in a real, dynamic Docker environment, the agent needs to autonomously navigate web pages, search for documents, and perform operations, rather than through offline evaluation on static datasets.

Weaknesses：
1. Lack of innovation: The idea of ​​proactive search and leveraging external knowledge bases is not new. On the contrary, many works have applied this approach, and a large number of agent frameworks have already used it, which makes the contribution of this paper quite limited. Proactive search: [1] Appagent: Multimodal agents as smartphone users and [2] Appagent-pro: A proactive gui agent system for multidomain information integration and user assistance, prior knowledge base: [3] Mobile-agent-e: Self-evolving mobile assistant for complex tasks.
2. Limitations of evaluation applicability: For agents with small parameters, there is no specific training for document retrieval, which makes them unfairly treated in the current evaluation.
3. Experimental limitations: The limited number of applications and baselines involved significantly reduces the applicability of the current conclusions.

---

> ### Author Rebuttal · Authors · 2026-03-31
>
> # Q1: Lack of innovation
>
> **A1**: We would like to clarify that the primary contribution of this work lies in the **formalization of a new interaction paradigm**, **Proactive Document-Guided Action**, which grounded in **interactive, open-web environments**.
>
> - Appagent [1,2] relies on **pre-collected** exploration logs or static UI descriptions, and Mobile-agent-e [3] focuses on **historical memory** retrieval. In contrast, DocOS evaluates the agent's ability to resolve **zero-shot tasks** in professional software by **autonomously seeking real-time information** from the open web (e.g., official online documentation).
> - While [1-3] often bake operational logic into the model or a local knowledge base, DocOS **decouples general reasoning from specific domain knowledge**. Our paradigm tests if an agent can "learn how to use an unseen tool" on the fly, which is essential for handling software updates and long-tail professional applications (e.g., IDEs, creative suites).
>
> By focusing on real-time, interactive information gathering, our work provides a methodological foundation for designing agents that can adapt to changing web content and integrate new knowledge without relying solely on pre-collected datasets.
>
> # Q2: Limitations of evaluation applicability
>
> **A2**：We would like to clarify **a potential factual misunderstanding** regarding the models' capabilities. The models evaluated in our study **possess inherent web-page understanding and navigating proficiency**:
>
> - **Qwen3-VL:** The sft data of this model explicitly "include, but are not limited to, spatial reasoning for embodied intelligence, **image-grounded reasoning for fine-grained visual understanding**, spatio-temporal grounding in videos for robust object tracking, and **the comprehension of long-context technical documents spanning hundreds of pages**."
> - **UI-TARS-1.5:** "we built a large-scale dataset comprising **screenshots and metadata from websites, apps, and operating systems**."
> - **MAI-UI**: "In the **Navigation Task** Generation stage, we leverage multiple sources (**APP manuals**, expert-designed tasks, and open-source data) to construct high-quality seed tasks.
>
> # Q3: Experimental limitations
>
> **A3**:
>
> 1. Scale of Applications
>
> We would like to clarify that 817 tasks in **DocOS** actually represent **a substantial and highly competitive scale** compared to existing high-level GUI benchmarks, such as **OSWorld (369 tasks)**,  **CRAB (100 tasks)**, and **WebArena (812 tasks)**.
>
> [1]Xie, T., Zhang, D., Chen, J., et al. Osworld: Benchmarking multimodal agents for open-ended tasks in real computer environments.
>
> [2] Xu, T., Chen, L., Wu, D.-J., et al. Crab: Cross-environment agent benchmark for multimodal language model agents.
>
> [3]Zhou, S., Xu, F. F., Zhu, H., et al. Webarena: A realistic web environment for building autonomous agents.
>
> 2. Baselines
>
> Regarding the baselines, we have added the larger-scale **Qwen-3** model and the **AutoGen** framework, and we have included the results in the final version.
>
> | Model | TUI   | HPP  | TCR  |
> | -- | -- | --- | --- |
> | Qwen3-VL-32B | 45.66 | 0.55 | 6.21 |
> | AutoGen(Qwen3-VL-8B) | 41.21 | 0.53 | 5.13 |
>
> # Q4: Classification criterion
>
> **A4**: We apologize for any misunderstanding and would like to clarify that the task difficulty is determined based on the standard steps in the **second-stage execution process**, rather than the number of steps in the first-stage retrieval process. Following **WebArena** and **VisualWebArena**, we adopt this classification criterion. Additionally, **OSWorld** uses execution time for categorization, which is also influenced by the number of steps, further supporting the reasonableness of our classification standard.
>
> # Q5: Innovation in document organization and data collection
>
> **A5**: We would like to clarify that this evaluation may overlook the core innovations of DocOS. As shown in Table 1, DocOS introduces a **formalization of a new interaction paradigm**, **Proactive Document-Guided Action**, grounded in **interactive, open-web environments**. Existing benchmarks, in contrast, do not support proactive document retrieval or document-guided execution. Furthermore, the dataset size is constrained by **a strict curation process**, which is the main factor limiting its expansion.
>
> # Q6: Document significance
>
> **A6:** We have addressed the reviewers' concerns through ablation studies presented in the paper.
>
> 1. **Section 5.1** explicitly compares the two settings, “with Document” and “without Document” in Table 4. The results show that all models achieve higher **TCR** scores when a document is provided.
> 2. **Section 5.2** further validates the upper bound of an **“ideal document”**: when ground-truth instructions are provided, **MAI-UI-8B** and **GELab-Zero-4B-preview** achieve improvements of 9.85% and 10.78%, respectively.
>
>  These results collectively **demonstrate the effectiveness of incorporating documents** in guiding task execution.

---

> > ### Author Rebuttal · Reviewer_w5N4 · 2026-04-02
> >
> > Thank you for the detailed rebuttal and clarifications. My questions have been largely addressed. However, despite the good task types and coverage of docos, it still offers limited innovation compared to previous work.
> >
> > Btw, these two questions are not resolved.
> >
> > (1) Q3 is asking whether these base models have good rag-doc performance, not your answer about page comprehension ability, etc?
> >
> > (2) Furthermore, Q6 is raised because the ablation study in sec 5 shows that the improvement is very limited, which is significantly different from previous work. Of course, this may be related to the fact that the current model has more pre-training experience?

---

> > > ### Author Response · Authors · 2026-04-08
> > >
> > > Thank you for your feedback. We would like to address the two remaining questions as follows:
> > >
> > > 1. Unlike traditional RAG-doc tasks, our task is more closely related to page understanding. This is because our evaluation focuses on models’ ability to perform **dynamic web retrieval**, which requires understanding and leveraging the structure and content of web pages, rather than solely retrieving static documents. Therefore, `RAG-doc performance and our task are not directly comparable`.
> > > 2. Our experimental results demonstrate **consistent and meaningful improvements** in Section 5,  which suggests that the improvements are robust and effectively translate to the proposed tasks.

---

### Official Review · Reviewer_rFAZ · 2026-03-13

**Soundness:** 3
**Presentation:** 3
**Significance:** 3
**Originality:** 3
**Overall Recommendation:** 4
**Confidence:** 3

**Summary:**

This paper clarifies the gap between inner model knowledge and the real-world task. The GUI agents should resort to more external info (like docs) to finish the task. Additionally, the authors provide a brand new benchmark DocOS, consisting of hundreds of tasks, which require external doc searching for task completion.

**Compliance With Llm Reviewing Policy:**

Affirmed.

**Final Justification:**

The rebuttal addresses my major concerns. I will keep my score.

**Key Questions For Authors:**

See above

**Strengths And Weaknesses:**

Strength:
1. The proposed insight (Agent requires external info) is practice in real-world application.
2. Insightful findings. e.g., some agent cannot retrieve key info, while others cannot process the key info.


Weakness:
1. You collect 817 data entries.In section 3.3, you argue that you sample 20% of the data and 92.50% of them satisfy the requirements. However, 817 \* 0.2 \* 0.9250 = 151.145, which indicates that your calculation results may contain some error.

2. You remove the irrelevant resources/elements to "prevent failures unrelated to agent performance". However, understanding complex webpage and retrieving critical data is also the key ability of GUI application.  In real-world scenarios, it's likely to encounter sudden events irrelevant to your goal. I think this filtering process may limit the application value of the proposed benchmark.

3. Repos: "condectedd" in section 4.2, "byGUI-Owl-7B" in section 5.1

4. The definition of POMDP better include the goal/goal state/verifier

---

> ### Author Rebuttal · Authors · 2026-03-31
>
> # Q1: Clarification on Task Sampling and Pass Rate
> **A1**：We thank the reviewers for their careful examination. There is a minor ambiguity in the wording, which we would like to clarify:
>
> The actual number of sampled tasks is **160** (approximately 19.58%, rounded to 20%), of which **148** passed verification, resulting in a pass rate of **92.50%** (148/160 = 0.925). We will revise the statement in Section 3.3 to: **“We sample approximately 20% (160 tasks) for human evaluation, finding 92.50% (148 tasks) satisfy the requirements.”**
> # Q2: Clarification on Document Filtering Procedure
> **A2**：We apologize for any misunderstanding and would like to clarify the timing of document filtering:
>
> - The filtering process occurs during the **data collection and processing stage**, aiming to remove irrelevant elements such as navigation bars and extract structured procedural knowledge from the raw webpages (step instructions, UI element descriptions). This does **not** affect dynamic interactions during the evaluation phase.
> - During the evaluation phase, agents interact with the **full browser environment and unfiltered search results**, requiring the ability to **understand complex webpages and extract key information from noisy content**.
>
> Additionally, we will clarify in Section 3.2: **“This preprocessing step is only used to construct the ground-truth document; the agent evaluation environment remains fully dynamic and unfiltered.”**
> # Q3: Typographical Corrections
> **A3**：We thank the reviewers for their careful proofreading. We will correct the following in the final version:
>
> - In Section 4.2, **“conductedd” → “conducted”**
> - In Section 5.1, **“byGUI-Owl-7B” → “by GUI-Owl-7B”**
> # Q4: Definition of POMDP
> **A4:** We thank the reviewer for this insightful suggestion. We will revise the POMDP definition in Section 3.1 as follows:
>
> $\mathcal{M} = \langle \mathcal{S}, \mathcal{A}, \mathcal{O}, \mathcal{T}, \mathcal{I}, \mathcal{G}, \mathcal{V} \rangle$
>
> with the additions:
>
> - $\mathcal{G}$: Goal conditions, defining the expected state for task completion.
> - $\mathcal{V}: \mathcal{S} \times \mathcal{G} \rightarrow \{0, 1\}$: A verifier that determines whether the current state satisfies the goal.
>
> We will also add the clarification: “Task success is determined by state-based verification $\mathcal{V}(s_{final}, \mathcal{G}) = 1$, rather than superficial metrics like exact URL matching.”

---

> > ### Author Rebuttal · Reviewer_rFAZ · 2026-04-04
> >
> > Thanks for your reply. I will keep the positive score, good luck~

---

### Official Review · Reviewer_bn9H · 2026-03-14

**Soundness:** 1
**Presentation:** 3
**Significance:** 3
**Originality:** 3
**Overall Recommendation:** 4
**Confidence:** 4

**Summary:**

This paper introduces DocOS, a new benchmark that challenges GUI agents to 1. Navigate the web and acquire relevant documentation, and 2. Use the retrieved documentation to solve GUI automation tasks. They study two aspects to show that GUI agents of today fail to navigate and retrieve the correct docs and even when provided with oracle documents, fail to properly ground their actions in the facts from the documents. The paper attempts to expose a critical weakness in GUI agents, specifically their inability to autonomously gather the necessary domain knowledge and utilize the domain knowledge to guide their actions.

**Compliance With Llm Reviewing Policy:**

Affirmed.

**Final Justification:**

The authors have resolved specific questions about motivation and evaluation that I asked. I am raising my score from 3 -> 4.

**Key Questions For Authors:**

See Weaknesses.

**Limitations:**

Yes.

**Strengths And Weaknesses:**

Strengths:
1. This is an important evaluation direction for computer use agents and has the potential to reveal important issues in continual learning abilities. Extended to the real world, this benchmark can provide early hints on whether a GUI agent can suitably complete tasks in novel and underrepresented applications.
2. The paper is clearly written, and it is very easy to follow through their arguments and reasoning.

Weaknesses:
1. I have an issue with the motivation for the pipeline used by this benchmark. It feels quite unnecessary for agents to literally browse the web using GUI navigation to reach the appropriate documents. This information can be easily obtained with a basic textual web search, which is a ubiquitous tool used by all agents of the day.

2. The second issue is with baselines considered by the authors. 5/6 models evaluated as GUI agents are specialist GUI agents that are trained to only solve GUI tasks given instructions. It makes sense that they will struggle to do knowledge retrieval and use that knowledge. These tasks need to be evaluated with generalist models that can do both to show that frontier models actually struggle at this task. Qwen3-VL-8B would be a good test, but then again, all of these models are the smaller variants of frontier models. This tells us that smaller models struggle at this task, but does not support the general claim in the paper that says: _Our results show that current agents are fundamentally limited by dual bottlenecks: inaccurate information localization during document search and unfaithful grounding of instructions into actions, underscoring a critical gap toward robust, self-guided deployment_. You need to test your benchmark on at least some frontier models like Kimi-K2, Larger variants of Qwen, Claude Sonnet, GPT-5, etc. Note that you don't need to test all these models, but to show that this benchmark is scalable and today's frontier agents actually do struggle on it.

3. The baselines don't include any agentic frameworks, just models. A task requiring Knowledge Retrieval + Execution feels like something agentic frameworks are designed for.

---

> ### Author Rebuttal · Authors · 2026-03-31
>
> # Q1: Motivation
> **A1**: We apologize for any confusion and would like to clarify that modern webpages are **interactive systems**, rather than **static text**. In this scenario, GUI-based navigation has practical advantages over basic text-based web search, particularly when handling interactive elements such as:
>
> - **multi-hop interactions** – GUI navigation allows agents to perform sequential, dependent actions (e.g., clicking through menus or tabs) to progressively narrow down and locate the precise information, which simple text search cannot replicate[1].
> - **Dynamic, JS-heavy sites:** Single Page Applications (SPAs) where data is loaded asynchronously[2].
> - **embedded media** – Instructions or information may be conveyed via images, videos, or interactive widgets; GUI-based interaction allows agents to access and interpret these media in context, beyond what textual retrieval can capture.
>
> Mainstream text-based web search approaches typically reduce complex webpages to simple document processing, bypassing the real webpage environment and the need for UI understanding. By choosing GUI navigation, **DocOS** can better handle these **interactive and hierarchical layers**, motivating the pipeline choice for more realistic web environments.
>
> [1]Zha Zhengjun, Zheng Xiaoju. Query and Feedback Technologies in Multimedia Information Retrieval[J]. Journal of Computer Research and Development, 2017, 54(6): 1267-1280. DOI: 10.7544/issn1000-1239.2017.20170004
>
> [2]Anchor. Browser Agent vs. LLM Agent: Why Reasoning Alone Breaks on the Real Web.
> # Q2: Baselines
> **A2**: Thank you for your suggestion. We evaluated DocOS on a larger-scale Qwen model. The experimental results show that this benchmark is scalable and further validate “current agents are fundamentally limited by dual bottlenecks: inaccurate information localization during document search and unfaithful grounding of instructions into actions, underscoring a critical gap toward robust, self-guided deployment.”
>
> | Model        | TUI   | HPP  | TCR  |
> | ------------ | ----- | ---- | ---- |
> | Qwen3-VL-32B | 45.66 | 0.55 | 6.21 |
> # Q3: Agentic Frameworks
> **A3**: Thank you for your suggestion. We have supplemented the evaluation experiments in the agent framework. The results indicate that incorporating the agent framework does not lead to significant improvements.
>
> | Model                | TUI   | HPP  | TCR  |
> | -------------------- | ----- | ---- | ---- |
> | AutoGen(Qwen3-VL-8B) | 41.21 | 0.53 | 5.13 |

---

> > ### Author Rebuttal · Reviewer_bn9H · 2026-04-03
> >
> > "modern webpages are interactive systems, rather than static text": This is true, and testing agents on navigating these interfaces is a useful task that has already been extensively studied through benchmarks like OSWorld, WebArena, etc. However, documentation pages for applications, including the applications you have in your benchmark, can be directly retrieved as static text and used to augment the context of agents. Are there any applications and corresponding documentation pages in your benchmark, where it is actually impossible to get some critical information from text-based retrieval?
> >
> > Qwen3-VL-32B is indeed a larger variant of Qwen-3-Vl-8B, but it is not near state-of-the-art models in terms of its GUI navigation, document parsing, or agentic abilities. If this is a compute budget limitation, I understand, but to make general claims that "current agents are fundamentally limited by dual bottlenecks..." you need to also evaluate State-of-the-art generalist models.
> >
> > Regarding the autogen result, "The results indicate that incorporating the agent framework does not lead to significant improvements". This is a very surprising result, but a substantial claim to make when you have only tested one framework against one model. I would like to follow up and ask about the experimental settings used in your implementation of Autogen.

---

> > > ### Author Response · Authors · 2026-04-08
> > >
> > > We thank the reviewer for the helpful feedback and address the points below.
> > >
> > > (1) To illustrate that some documents in our benchmark contain key information that cannot be obtained through text retrieval alone, we provide the following example:
> > >
> > > On the webpage https://www.jetbrains.com/help/pycharm/2026.1/cleaning-system-cache.html, the textual description “`select the relevant action`” is ambiguous. However, the webpage includes images that clearly show specific options such as “`Clear file system cache and Local History`” and “`Clear VCS Log caches and indexes`.” This demonstrates that interactive and dynamic retrieval is necessary.
> > >
> > > (2) We acknowledge that Qwen3-VL-32B is not state-of-the-art. It was chosen for reasons of computational efficiency and reproducibility. We have explicitly limited our conclusions to the evaluated models and settings in the final version.
> > >
> > > (3) In our Autogen setup, we use a **single-agent configuration**, where the agent’s actions are wrapped as **MCP**. Other settings, such as context window size and history length, are **kept consistent with our other experiments**.

---

### Decision · Program_Chairs · 2026-04-30

**Decision:**

Accept (regular)

**Comment:**

DocOS is an evaluation benchmark for evaluating computer use agents. Unlike OSWorld, the main contribution is that these tasks require fetching relevant documentation on how to do them. The benchmark is human-curated and verified, and evaluation is through state-based verification in a sandbox.

Overall, all reviewers felt that the paper is strong on these aspects:
1. Without non-parametric knowledge, these tasks are hard. Most models perform close to 0.
2. The largest model they evaluated (Qwen3-VL-32B) achieves only 6%. Even at the retrieval stage, agents seem to fail.
3. Collecting such data is non-trivial.

Unfortunately, no results with frontier models are presented. It would be important to validate if this dataset still has runway.